# Microbial enzymes induce colitis by reactivating triclosan in the mouse gastrointestinal tract

Jianan Zhang [1,17], Morgan E. Walker [2,17], Katherine Z. Sanidad[1,17], Hongna Zhang[3,4,17], Yanshan Liang[3], Ermin Zhao[1], Katherine Chacon-Vargas[1], Vladimir Yeliseyev[5], Julie Parsonnet [6], Thomas D. Haggerty[6], Guangqiang Wang[1,7], Joshua B. Simpson[2], Parth B. Jariwala[2], Violet V. Beaty[2], Jun Yang [8], Haixia Yang[1], Anand Panigrahy[1], Lisa M. Minter[9], Daeyoung Kim[10], John G. Gibbons [1], LinShu Liu[11], Zhengze Li[1], Hang Xiao [1], Valentina Borlandelli [12], Hermen S. Overkleeft[12], Erica W. Cloer[13], Michael B. Major[14], Dennis Goldfarb [15], Zongwei Cai [3✉], Matthew R. Redinbo [2✉] & Guodong Zhang [1,16✉]

Emerging research supports that triclosan (TCS), an antimicrobial agent found in thousands of consumer products, exacerbates colitis and colitis-associated colorectal tumorigenesis in animal models. While the intestinal toxicities of TCS require the presence of gut microbiota, the molecular mechanisms involved have not been defined. Here we show that intestinal commensal microbes mediate metabolic activation of TCS in the colon and drive its gut toxicology. Using a range of in vitro, ex vivo, and in vivo approaches, we identify specific microbial β-glucuronidase (GUS) enzymes involved and pinpoint molecular motifs required to metabolically activate TCS in the gut. Finally, we show that targeted inhibition of bacterial GUS enzymes abolishes the colitis-promoting effects of TCS, supporting an essential role of specific microbial proteins in TCS toxicity. Together, our results define a mechanism by which intestinal microbes contribute to the metabolic activation and gut toxicity of TCS, and highlight the importance of considering the contributions of the gut microbiota in evaluating the toxic potential of environmental chemicals.

[1] Department of Food Science, University of Massachusetts, Amherst, MA, USA. [2] Departments of Chemistry, Biochemistry, Microbiology and Genomics, University of North Carolina at Chapel Hill, Chapel Hill, NC, USA. [3] State Key Laboratory of Environmental and Biological Analysis, Department of Chemistry, Hong Kong Baptist University, Hong Kong, SAR, China. [4] Department of Occupational and Environmental Health, School of Public Health, Qingdao University, Qingdao, China. [5] Massachusetts Host-Microbiota Center, Department of Pathology, Brigham and Women's Hospital, Boston, MA, USA. [6] Department of Medicine and Department of Health Research and Policy, Stanford University, Stanford, CA, USA. [7] School of Medical Instrument and Food Engineering, University of Shanghai for Science and Technology, Shanghai, China. [8] Department of Entomology and Nematology, University of California, Davis, CA, USA. [9] Department of Veterinary & Animal Sciences, University of Massachusetts, Amherst, MA, USA. [10] Department of Mathematics and Statistics, University of Massachusetts, Amherst, MA, USA. [11] Eastern Regional Research Center, Agricultural Research Service, United States Department of Agriculture, Wyndmoor, PA, USA. [12] Department of Bioorganic Synthesis, Leiden Institute of Chemistry, Leiden University, Leiden, Netherlands. [13] Lineberger Comprehensive Cancer Center, University of North Carolina at Chapel Hill, Chapel Hill, NC, USA. [14] Department of Cell Biology and Physiology, and Department of Otolaryngology, Washington University, St. Louis, MO, USA. [15] Department of Cell Biology and Physiology, Institute for Informatics, Washington University, St. Louis, MO, USA. [16] Department of Food Science and Technology, National University of Singapore, Singapore, Singapore. [17]These authors contributed equally: Jianan Zhang, Morgan E. Walker, Katherine Z. Sanidad, Hongna Zhang. ✉email: zwcai@hkbu.edu.hk; redinbo@unc.edu; zhanggd@nus.edu.sg

The incidence and prevalence of inflammatory bowel disease (IBD), the chronic inflammation of intestinal tissues[1], have risen dramatically in recent decades[2]. In 2015, an estimated ~1.3% of U.S. adults (~3 million) were diagnosed with IBD[3], representing a 50% increase from 1999 (~2 million)[4]. IBD can severely impact life quality, as symptoms include abdominal pain, diarrhea, and rectal bleeding, and there is no cure. Current anti-IBD treatments can result in serious side effects and idiopathic patient responses[5]. More alarmingly, IBD patients have increased risks of developing colorectal cancer[6]. While the increase in IBD prevalence has been linked to exposure to environmental chemicals[7–9], the potential mechanisms involved have remained unclear.

Triclosan (TCS) is an antimicrobial ingredient present in more than 2000 consumer and industrial products and detected in ~75% of urine samples tested in USA[10]. In 2016, the USA Food and Drug Administration (FDA) removed TCS from over-the-counter handwashing products; however, it remains approved for use in a wide range of products such as toothpaste, mouthwash, hand sanitizers, cosmetics, and toys[10,11]. It is a ubiquitous environmental contaminant and a top-ten US river pollutant[10]. Health problems connected to TCS include increased risks of allergies and asthma, altered immune responses, disruption of endocrine functions, and increased development of antibiotic resistance[10]. Specific to the gastrointestinal tract, we recently showed that exposure to TCS, at human-relevant doses, increased the severity of colitis and exaggerated the development of colitis-associated colorectal cancer in mouse models[12]. This finding supports that TCS could be a potential risk factor for IBD and associated diseases, though further studies are needed to determine its impacts in human populations[12,13].

Because TCS exposure fails to promote colonic inflammation in germ-free mice, the toxicity of TCS requires the presence of the gut microbiota[12]. However, the specific mechanisms connecting the gut microbiota with TCS toxicity are unknown. The discovery of such gut microbial factors will define a molecular mechanism of TCS-driven gut pathology and increase our understanding of host-microbiota interactions. Previous studies have shown that once TCS enters the body, it is rapidly metabolized in host tissues, such as the liver, to form the glucuronide-conjugated metabolite TCS-glucuronide (TCS-G), which is biologically inactive and is thought to be quickly eliminated from the body[10,14]. Given this rapid metabolic inactivation, though, it has remained unclear how exposure to low-dose TCS causes gut toxicity in vivo. We hypothesize that gut microbial enzymes act on key TCS metabolites in the colon, leading to unique gut metabolic profiles highlighted by reactivation of TCS in the gut and resulting in subsequent gut toxicology. Here, using a range of in vitro, ex vivo, and in vivo approaches, we define the unique TCS metabolic patterns that lead to intestinal toxicity, pinpoint the specific gut microbial enzymes that drive the colitis-promoting effects of TCS, and disrupt those enzymes by selectively targeting the gut microbiota. These data reveal the critical roles played by gut microbial enzymes in the metabolic activation and gut toxicity of TCS.

## Results

**Unique TCS metabolic profile in the mouse gut**. We first sought to determine whether the lower intestinal tract exhibited a TCS metabolic profile distinct from other tissues. We treated mice with 80 ppm TCS via diet for 4 weeks, then employed LC-MS/MS to analyze the concentrations of TCS and its metabolites in a range of mouse tissues. We determined the TCS treatment scheme to model human exposure to TCS based on our previous study[12]. We found that after TCS exposure, the dominant TCS compound found in the liver, bile, heart, and small intestine was the biologically inactive conjugated metabolite TCS-G. In contrast, the mouse cecum and colon were dominated by free TCS. The TCS metabolites found in the digesta of the 4th section of the mouse small intestine were 36.9% free TCS, 55.4% TCS-G, and 7.7% sulfate-conjugated TCS (TCS-sulfate); while the fecal content exhibited 99.1% free TCS with only 0.7% TCS-G and 0.2% TCS-sulfate (Fig. 1a). These results show that the colon has a distinct metabolic profile of TCS compared to other tissues and uniquely contains nearly universally free TCS.

To further validate this finding, we treated mice with lower doses of TCS and analyzed its metabolic profile in colon tissues. We treated mice with 1, 10, and 80 ppm TCS via diet for 4 weeks, and found that at all tested doses, the gut tissues had similar metabolic profiles of TCS and were characterized by a high abundance of free TCS: ~94-100% of detected TCS species in gut sections, including the colon digesta, cecum digesta, and feces, were present as free TCS (Fig. 1b); while a mixture of TCS and its metabolites is observed elsewhere (Supplementary Fig. S1). In addition, LC-MS/MS showed a dose-dependent effect of TCS exposure on the gut concentration of TCS: the concentrations of free TCS in the colon digesta were 1.5, 14.7, and 92.2 pmol/mg tissue after exposure to 1, 10, and 80 ppm TCS in diet, respectively (Fig. 1b). These findings further support that the colon has a unique TCS metabolic profile and contains a high abundance of free TCS.

**Unique TCS metabolic profile in the human gut**. To broaden our understanding of TCS metabolic profiles in the gut, we next analyzed TCS metabolism in human subjects. We utilized urine and stool samples from a previous study in which the recruited human subjects were first subjected to a washout period (no usage of TCS-containing products), then were randomly assigned to two groups that used personal care products like toothpaste with or without TCS for up to four months (see the scheme of the experiment in Fig. 1c)[15]. Previous data had established that humans are primarily exposed to TCS via toothpaste that provides a direct oral route to the gastrointestinal tract[16,17].

First, we tested whether TCS and its metabolites could be detected in stool or urine samples after using TCS-containing products. After the washout period, as expected, we found that most human subjects had very low levels of TCS at the beginning of the study ($t = 0$); although two subjects (one in the control group and one in the TCS group) showed detectable levels of TCS even at $t = 0$ (Supplementary Fig. S2a), likely due to ubiquitous nature of TCS in the environment[18,19]. LC-MS/MS showed that after even 1-month usage of TCS-containing products, TCS and its metabolites were detected in the urine and stool samples of TCS-exposed subjects but not in control subjects using TCS-free products (see stool analysis in Supplementary Fig. S2a and complete LC-MS/MS analysis data of stool and urine samples in Supplementary Tables S1 and 2).

Next, we analyzed metabolic profiles of TCS in TCS-exposed human subjects. LC-MS/MS showed that in all tested TCS-exposed subjects, the dominant compound in the stool samples was free TCS, while the dominant compound in the urine samples was TCS-G (Fig. 1d). The average molar concentration ratios of TCS, TCS-G, and TCS-sulfate were 99.2%:0.8%:0.0% in human stool vs. 1.6%:98.4%:0.0% in urine (Fig. 1d and Supplementary Fig. S2b, c). The concentrations of free TCS measured in stool were high, reaching up to ~1 pmol/mg tissue, which equates to ~1 μM. By contrast, TCS concentrations in urine were in the low-nM range (Supplementary Tables S1 and 2). Taken together, these results establish that the human gut exhibits a unique TCS metabolic profile compared to that found in the urine and contains a high abundance of free TCS.

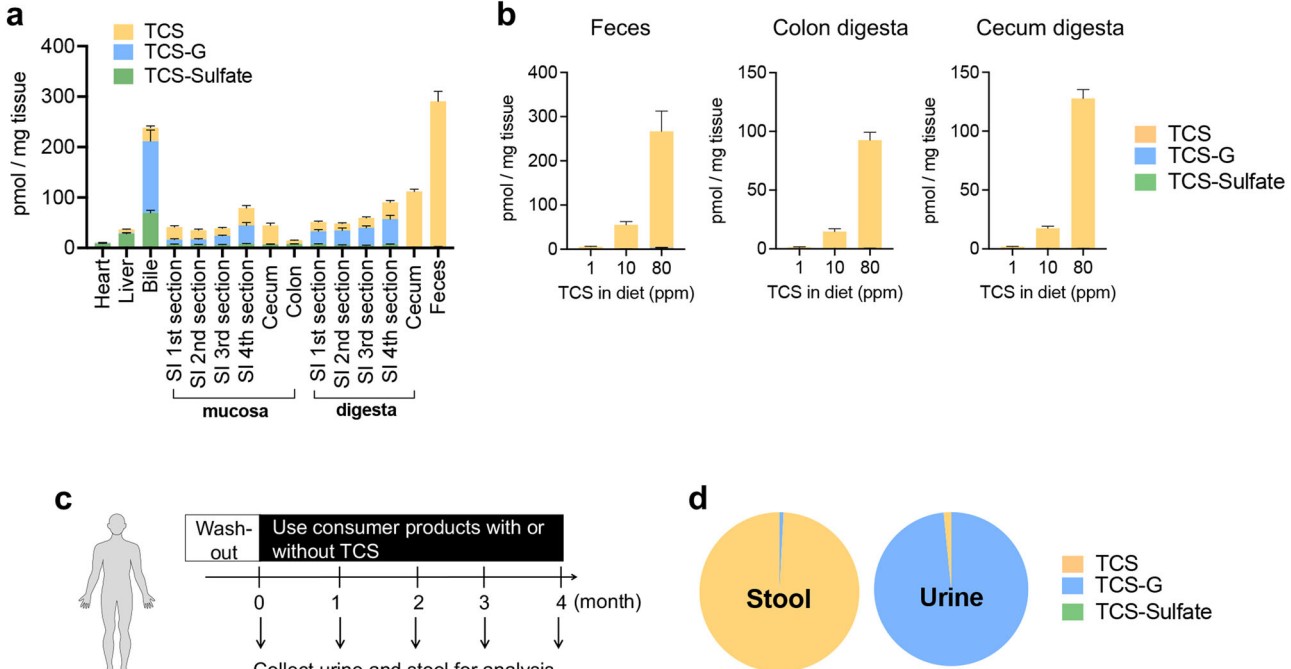

**Fig. 1 TCS exposure in mice and humans leads to accumulation of free TCS in the colon. a** After the mice were treated with 80 ppm TCS via diet for 4 weeks, cecum digesta and fecal contents exhibit high levels of free TCS while a mixture of TCS and metabolites is observed elsewhere ($n = 10$ mice per group). **b** Mice were treated with 1, 10, and 80 ppm TCS via diet for 4 weeks. At all test doses, the gut tissues, including feces, colon digesta, and cecum digesta, exhibit high levels of free TCS and low levels of its metabolites ($n = 7$–8 mice per group). **c** After a washout period, human subjects used personal care products, with or without TCS, for up to 4 months. Twenty-three fecal but only seven urine samples were collected. **d** The dominant compound in the stool samples of TCS-exposed human subjects was free TCS ($n = 23$), while the dominant compound in the urine samples of TCS-exposed human subjects was TCS-G ($n = 7$). See absolute concentrations of TCS and its metabolites in Supplementary Tables S1and 2. The data are mean ± SEM. Source data are provided with this paper. Abbreviations: TCS: triclosan, TCS-G: triclosan-glucuronide, TCS-sulfate: triclosan-sulfate. Part of the picture was adapted from motifolio.com.

**Gut microbiota converts TCS-G to TCS in the colon in vitro and in vivo.** The data presented above revealed that the concentration of TCS increased, while the concentration of TCS-G decreased, from the proximal to the distal regions of the intestinal tract (Fig. 1a). Thus, we hypothesized that gut microbiota participates in the conversion of TCS-G to TCS, leading to the accumulation of TCS in the lower gastrointestinal tract (see scheme in Supplementary Fig. S3). To test this hypothesis, we used a combination of approaches including in vitro culturing of gut bacteria, antibiotic-mediated suppression of gut bacteria in vivo, and germ-free mice to examine the roles of the gut microbiota in colonic metabolism of TCS.

First, we cultured gut bacteria under anaerobic conditions and tested whether the cultured bacteria can convert TCS-G to TCS in vitro. We found that fecal bacteria from both mice and humans were able to catalyze the conversion of TCS-G to TCS at levels significantly higher than control (Fig. 2a). These results support the conclusion that anaerobically cultured gut bacteria can catalyze the de-glucuronidation of TCS-G to create TCS.

Next, to test whether gut microbiota participates in the conversion of TCS-G to TCS in the colon in vivo, we examined whether antibiotic suppression of gut bacteria would alter the concentrations of TCS vs. TCS-G in colon digesta (see the scheme of the experiment in Fig. 2b). We employed an antibiotic cocktail from previous studies[20,21], having shown that this cocktail effectively reduced gut bacteria in mice[22,23]. To further validate this, we analyzed total fecal microbial biomass using the *16S rRNA* gene as a marker[21]. In agreement with previous studies by us and others[20–23], the cocktail employed caused a dramatic reduction of fecal bacteria in mice (Supplementary Fig. S4). Next,

LC-MS/MS studies showed that antibiotic treatment significantly reduced the concentration of free TCS while increasing by sixfold the concentrations of TCS-G in the fecal content (Fig. 2c). These results support the conclusion that gut bacteria contribute to the colonic conversion of TCS-G to TCS in vivo.

To further examine the roles of gut bacteria in the colonic metabolism of TCS, we tested the time-dependence of the antibiotic effects. Mice were pre-treated with or without the antibiotic cocktail, then received TCS via a one-time oral gavage, after which the metabolic profile of TCS was examined at $t = 4$, 8, 12, and 24 h (see the scheme of the experiment in Fig. 2d). We found that antibiotic suppression of gut bacteria reduced TCS and increased TCS-G in the colon digesta of mice in a time-dependent manner. Area under curve (AUC) analysis showed that within the 24-h period, antibiotic treatment reduced TCS by ~40%, while increasing TCS-G in the colon digesta by ~200-fold (Fig. 2e, f). This finding is consistent with the results presented above (Fig. 2c) and further supports the conclusion that gut bacteria contribute to the conversion of TCS-G to TCS in the colon.

Finally, we used germ-free mouse models to further examine the roles of gut microbiota in the colonic metabolism of TCS. We treated conventional mice or germ-free mice (established on C57BL/6 background) with TCS via a one-time oral gavage of TCS, then analyzed colonic TCS metabolic profiles at $t = 4$ and 8 h (see the scheme of the experiment in Fig. 2g). The time points of 4 and 8 h were determined based on our time-course study above (Fig. 2d–f). Compared with the conventional mice, germ-free mice exhibited reduced TCS and increased TCS-G in their colon digesta (Fig. 2h), consistent with results from the antibiotic

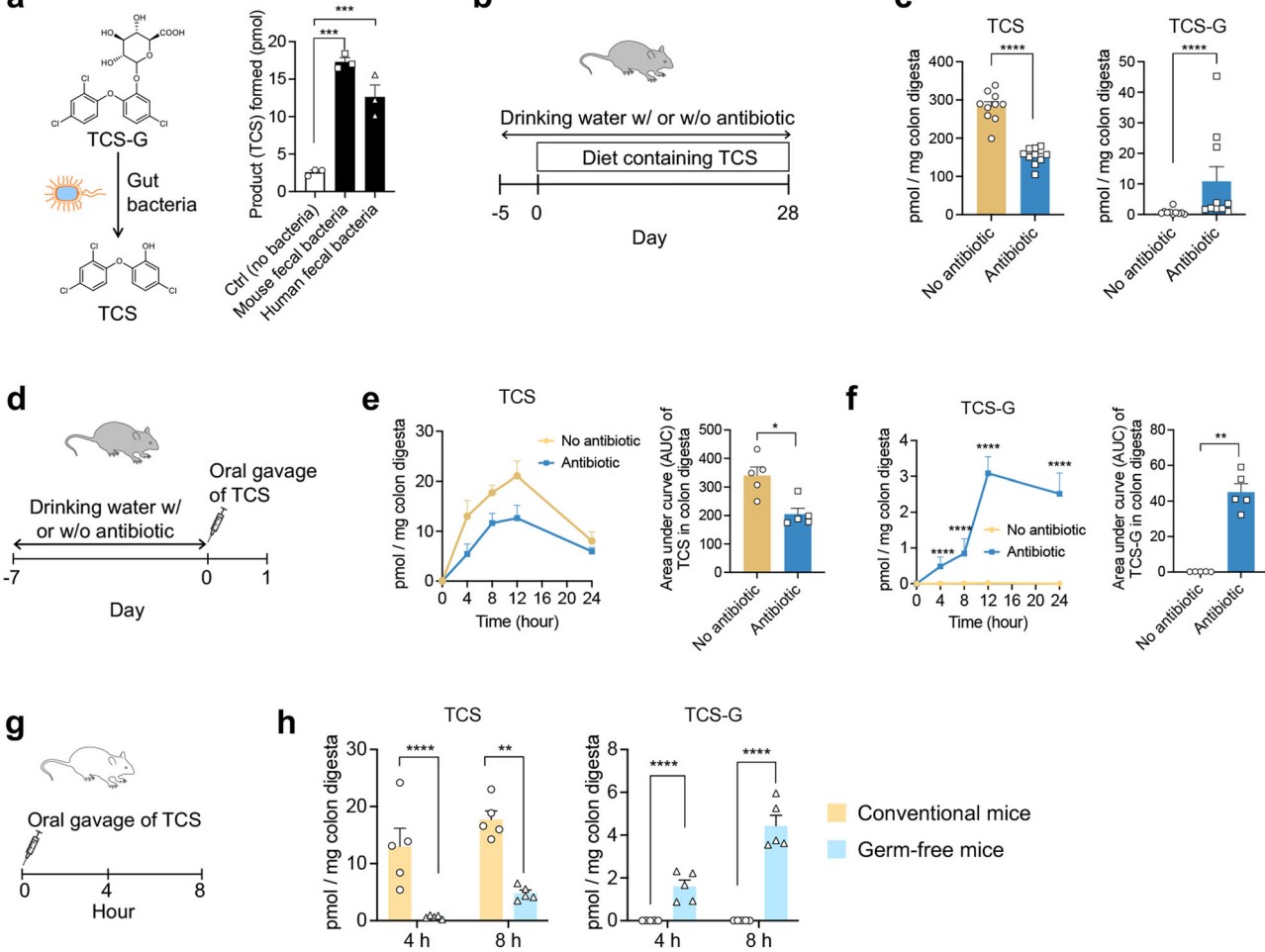

**Fig. 2 Gut bacteria convert TCS-G to TCS in vitro and in vivo. a** Fecal bacteria from mice and humans convert TCS-G to TCS in vitro ($n = 3$ per group). **b** C57BL/6 mice were treated with 80 ppm TCS via diet, with or without an antibiotic cocktail in drinking water, for 4 weeks. **c** Antibiotic treatment reduced TCS and increased TCS-G in fecal content of the mice ($n = 10$ mice per group). **d** C57BL/6 mice were pre-treated with or without antibiotics for 7 days and then dosed with a one-time oral gavage of 8 mg/kg TCS. **e, f** Antibiotic treatment reduced TCS and increased TCS-G in colon digesta of mice in a time-dependent manner. Left: time-course change in colon digesta ($n = 5$ mice per group for each time point). Right: area under curve (AUC) analysis. **g** Conventional or germ-free C57BL/6 mice were treated with a one-time oral gavage of 8 mg/kg TCS. **h** Compared with conventional mice, germ-free mice had reduced TCS and increased TCS-G in colon digesta ($n = 5$ mice per group for each time point). The data are mean ± SEM. To compare the two groups, Shapiro–Wilk test was used to verify the normality of data; when data were normally distributed, statistical significance was determined using two-sided *t* test; otherwise, significance was determined by Wilcoxon–Mann–Whitney test. *$P < 0.05$, **$P < 0.01$, ***$P < 0.001$, ****$P < 0.0001$. Source data are provided with this paper. TCS triclosan, TCS-G triclosan-glucuronide, TCS-sulfate triclosan-sulfate. Part of the picture was adapted from motifolio.com.

experiments (Fig. 2b–f). To further validate this finding, we compared TCS colonic metabolism in germ-free vs. conventional mice of a different strain, Swiss Webster. The concentration of TCS was reduced while the concentration of TCS-G was increased in colon digesta of the germ-free Swiss Webster mice compared to conventional animals (Supplementary Fig. S5). We observed the presence of free TCS in the colon of germ-free mice (Fig. 2h), and this could be from ingested TCS from the food: we showed that after mice were exposed to 80 ppm TCS in diet, part of the ingested TCS remained unchanged in the small intestine as free TCS was detected in the digesta of the small intestine (Fig. 1a). This could also happen in the germ-free mice and the free TCS in the small intestine could then enter the colon with the flow of digesta. Taken together, the results from in vitro culturing studies of fecal bacteria, antibiotic-mediated suppression of gut bacteria in vivo, and germ-free mouse models support the conclusion that commensal microbes convert TCS-G to TCS in the colon.

**Specific gut microbial β-glucuronidase (GUS) orthologs convert TCS-G to TCS.** We next sought to connect specific gut microbial enzyme(s) with the conversion of TCS-G to TCS. Because intestinal β-glucuronidase (GUS) enzymes have been shown to convert a wide range of glucuronidated metabolites to their corresponding aglycones[24–31], we hypothesize that gut microbial GUS orthologs would catalyze the conversion of TSC-G to TCS. The human and mouse gut microbiome have been shown to contain hundreds of unique gut microbial GUS enzymes, which exhibit distinct substrate specificities toward varying glucuronides[27,32]. Previous studies have shown that microbial GUS enzymes can be categorized into seven distinct clades based on active site architecture and/or cofactor binding[33]. We created a panel of 32 purified gut microbial GUS enzymes, which represent the seven clades, for in vitro enzymatic screening[29,30,34]. We first screened this panel for TCS-G cleavage activity using a coupled assay and found that Loop 1 and flavin mononucleotide (FMN)-binding GUS orthologs were most efficient at utilizing this

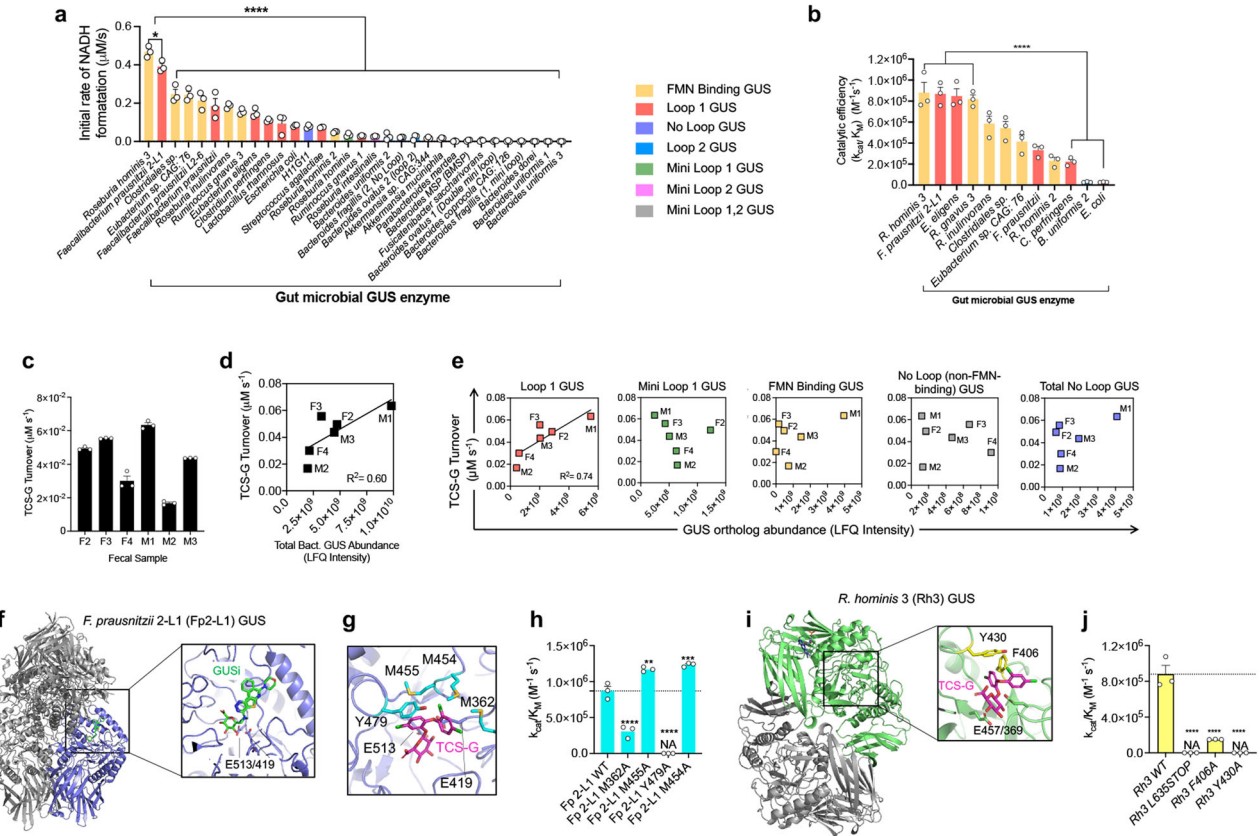

**Fig. 3 Specific gut microbial glucuronidase enzymes convert TCS-G to TCS. a** Screening a panel of 32 purified gut microbial β-glucuronidase (GUS) proteins representing seven structural clades using a coupled assay reveals that Loop 1 and FMN-binding GUS orthologs efficiently convert TCS-G to TCS in vitro. **b** Catalytic efficiency values determined by HPLC further indicate that Loop 1 and FMN-binding GUS orthologs show high TCS-G to TCS-conversion activities in vitro. **c** Enzymes extracted from human fecal samples exhibit variable TCS-G to TCS turnover rates ex vivo. **d** The abundance of total bacterial GUS enzymes identified in human fecal samples by activity-based probe-enabled proteomics is correlated with TCS-G turnover rate. **e** The abundance of Loop 1 GUS proteins identified in human fecal samples by activity-based probe-enabled proteomics, but not other types of GUS, is correlated with TCS-G turnover rate. **f** Crystal structure of *F. prausnitzii* 2-L1 (Fp2-L1) GUS with the overall enzyme tetramer shown (purple, gray) and a close-up of the GUSi-glucuronic acid conjugate (green) bound at the enzyme's active site with the catalytic glutamates highlighted. **g** TCS-G docked into the active site of Fp2-L1 GUS with residues selected for mutagenesis studies highlighted in cyan. **h** Catalytic efficiency values of wild-type (WT) and Fp2-L1 GUS mutant proteins. **i** Crystal structure of Rh3 GUS dimer (green, gray) with FMN bound (highlighted in blue in top monomer). Inset: TCS-G (magenta) was docked using Schrödinger and was found proximal to Y430 and F406 (yellow) at the Rh3 GUS active site (catalytic residues in green). **j** Catalytic efficiency values for Rh3 GUS mutants showing that the C-terminal domain, Y430 and F406 are important for TCS-G processing. The data are mean ± SEM, $n = 3$ biological replicates. All statistics were calculated using one-way ANOVA with Tukey's multiple comparisons test. *$P < 0.05$, **$P < 0.01$, ***$P < 0.001$, ****$P < 0.0001$. NA no activity. Source data are provided with this paper. TCS triclosan, TCS-G triclosan-glucuronide, GUS β-glucuronidase.

substrate (Fig. 3a). We then determined the catalytic efficiencies of TCS-G to TCS conversion by high-performance liquid chromatography (HPLC) for a select set of 12 GUS enzymes focused on Loop 1 and FMN-binding orthologs. Consistent with the results from the coupled assay, the more rigorous catalytic efficiency values showed that specific Loop 1 and FMN-binding GUSs were most effective at converting TCS-G to TCS in vitro (Fig. 3b).

We next employed a recently developed activity-based probe-enabled proteomics approach to provide an orthogonal measure of gut microbial GUS enzymes capable of processing TCS-G[35]. First, we examined human feces for their ability to activate TCS from TCS-G. Proteins were extracted from the fecal samples of three female and three male donors as described[35]. TCS-G turnover by the resultant mixtures revealed that all samples performed the reaction and that turnover rates varied by more than threefold (Fig. 3c). Next, the composition of GUS enzymes in each sample was determined using an activity-based covalent probe mimicking glucuronic acid linked to an affinity label, allowing enrichment of GUS proteins and their subsequent

proteomic identification and quantification, as described[35,36]. Total GUS abundance correlated with TCS-G turnover ($R^2 = 0.60$; Fig. 3d), as did Loop 1 GUS abundance ($R^2 = 0.74$), while the abundance of the other forms of GUS detected failed to exhibit correlation with TCS activation, including the abundance of FMN-binding GUS enzymes (Fig. 3e). The relative GUS composition of each fecal sample reveals that all contain Loop 1 GUS enzymes, the isoforms whose abundance correlated best with TCS-G turnover (Fig. 3e). Thus, taken together, the in vitro activity and coupled proteomic data support the conclusion that Loop 1 gut microbial GUS enzymes appear to be important drivers of TCS-G processing.

**Unique structural motifs are required for gut microbial GUS enzymes to process TCS-G.** We next examined the structural basis for efficient TCS-G cleavage by Loop 1 gut microbial GUS enzymes. We focused on *F. prausnitzii* 2-L1 (Fp2-L1) GUS, which was the most active Loop 1 GUS protein identified from our in vitro enzymatic assays (Fig. 3a, b). The crystal structure of Fp2-

L1 GUS was determined and refined to 2.2 Å resolution (Supplementary Table S3) and reveals a protein tetramer with a GUS inhibitor UNC10201652 (GUSi) covalently linked to glucuronic acid in each active site[24,37] (Fig. 3f). Fp2-L1 GUS was crystallized in the presence of GUSi and the reporter substrate p-nitrophenyl-glucuronide. GUSi has been shown to intercept the GUS catalytic cycle and produce the covalent GUSi-glucuronic acid adduct observed here and described previously[37]. GUSi adopted a similar binding conformation to that seen previously (PDB 6CXS) (Supplementary Fig. S6)[29]. Using the Schrödinger molecular modeling suite, we docked TCS-G into the active site of Fp2-L1 GUS and found that Y479 and three methionines (M454, 455, and 362) are positioned to potentially contact TCS-G (Fig. 3g). Mutation of Y479 or M362 to alanine significantly reduced TCS-G processing, while mutation of M454 or M455 to alanine significantly increased TCS-G processing, perhaps by reducing steric occlusion during TCS-G turnover (Fig. 3h). We confirmed by circular dichroism that the mutant proteins did not exhibit significant structural changes compared to the wild-type enzyme, indicating that the mutations are directly responsible for the observed loss of activity with TCS-G (Supplementary Fig. S7a, b). Furthermore, methionines 362 and 455 are unique to Fp2-L1 GUS compared to another Loop 1 GUS enzyme, E. coli GUS, that we show poorly utilizes TCS-G as a substrate (Fig. 3a, b, see the alignment of Fp2-L1 GUS and E. coli GUS in Supplementary Fig. S8). Together, these results demonstrate that specific Fp2-L1 GUS residues are important for TCS-G processing. Finally, it is likely that the loop structure of each Loop 1 GUS enzyme plays a key role in substrate processing ability. Unfortunately, this loop remains unresolved in several of the structures resolved to date, making it difficult to elucidate the structural role that this loop plays in substrate recognition. A multiple sequence alignment reveals that there is little sequence identity between the Loop 1 GUS enzymes (Supplementary Fig. S9). For example, even for enzymes that have similar catalytic efficiencies, like E. eligens and Fp2-L1 GUS, there are few commonalities in their Loop 1 regions that would allow for correlations to be made between loop structure and enzyme function (Supplementary Fig. S9). Nonetheless, it is still apparent that the presence of a loop at the Loop 1 position appears to be favorable for TCS-G binding when compared to other loop classes.

In addition to Loop 1 GUS enzymes, our in vitro results with purified enzymes showed that FMN-binding GUS proteins, notably R. hominis 3 (Rh3) GUS, also efficiently process TCS-G (Fig. 3a, b). Thus, we determined the crystal structure of Rh3 GUS and refined it to 2.4 Å resolution (Supplementary Table S3). The structure reveals a protein dimer with solvent-accessible active sites located ~30 Å from the bound FMN molecules (Fig. 3i). Using analogous docking and mutagenesis methods to those outlined above, we validated that specific structural motif, including residues F406 and Y430, as well as the C-terminal domain (see the scheme of the C-terminal domain in Supplementary Fig. S10), are critical for processing TCS-G (Fig. 3j). Again, these mutant proteins exhibited no structural changes when compared to the wild-type protein (Supplementary Fig. S7c, d). F406 is unique to Rh3 GUS compared to another FMN-binding GUS protein, R. hominis 2 (Rh2) GUS, that we show poorly processes TCS-G (Fig. 3a, b, see the alignment of Rh3 GUS and Rh2 GUS in Supplementary Fig. S11). Rh3 GUS and Rh2 GUS also vary in their C-terminal regions, as their C-terminal sequences share only 27% sequence identity, likely contributing to their differences in activity. Taken together, these structural studies reveal that gut microbial GUS enzymes that show efficient TCS reactivation activities contain motifs that are unique to these orthologs compared to other enzymes that do not effectively utilize TCS-G as a substrate.

**Targeted inhibition of gut microbial GUS abolishes the colitis-promoting effects of TCS in vivo.** We determined the extent to which the targeted inhibition of gut microbial GUS enzymes impacts the gut toxicity of TCS in vivo. Because genetic tools that specifically target gut microbial enzymes are sparse[30,38], we used a pharmacological approach and employed the GUS inhibitor UNC10201652[24,37] (GUSi; see the binding of GUSi in the active site of Fp2-L1 GUS in Fig. 3f–g and the chemical structure of GUSi in Fig. 4a). First, we tested the effect of GUSi on TCS-G processing in vitro and found that it inhibited the conversion of TCS-G to TCS by purified Fp2-L1 GUS enzyme, as well as several other Loop 1 GUS enzymes, in a dose-dependent manner, with IC$_{50}$ values of 0.64–4.9 μM (Fig. 4b). We were surprised to find that GUSi also inhibited, albeit with less potent IC$_{50}$ values of 3.7–13 μM, the processing of TCS-G by FMN-binding GUS enzymes (Fig. 4c). Previous data on GUSi had indicated that this compound was most efficacious against Loop 1 GUS enzymes[24,29,37]. Thus, the data here show that this chemotype exhibits the ability to inhibit FMN-binding gut microbial GUS enzymes as well. Next, we tested the effect of GUSi on TCS-G processing by the fecal enzyme mixtures ex vivo. While only two male and two female fecal samples remained at this stage for testing, we found that GUSi inhibited TCS-G processing in a manner that reflected the GUS levels present in each sample examined. In particular, GUSi exhibited more effective inhibition of ex vivo samples containing higher levels of Loop 1 GUS enzymes (Fig. 4d, e). These data establish that the GUSi blocks TCS-G processing in vitro and in human fecal extracts ex vivo, with effects on both Loop 1 and FMN-binding GUS enzymes.

After demonstrating that GUSi inhibits GUS-mediated TCS-G processing, we further characterized GUSi. Our previous study showed that GUSi has no effect on growth of E. coli or on the activity of mammalian GUS enzyme; deficiency of human GUS results in Sly Syndrome, a potentially fatal lysosomal storage disease[39]. In addition, we showed that GUSi has no effect on the proliferation of epithelial cells in the ileum, proximal or distal colon of the treated mice[29]. Here we further studied its effects on gut physiology. First, we treated C57BL/6 mice with 1 mg/kg GUSi via oral gavage (a treatment scheme determined from our previous studies[29,30]) and found that a 3- to 4-week treatment with GUSi had little effects on body weight, colon length, colonic or systematic inflammation, or colon histology in mice (Supplementary Fig. S12). GUSi treatment also had little effect on the diversity or composition of fecal microbiota in mice (Supplementary Fig. S13). Next, we found that a 24-h treatment with GUSi, at a concentration up to 10 μM, had little effect on the growth of mouse or human intestinal cells in vitro (Supplementary Fig. S14). Taken together, these results demonstrate that GUSi effectively inhibited GUS-mediated TCS-G processing, with little effect on commensal microbes, mammalian intestinal cells, or mammalian GUS enzyme, supporting that GUSi is highly selective toward the gut microbial GUS enzymes and therefore it is feasible to use GUSi to study the functional roles of microbial GUS enzymes in the gut toxicity of TCS.

We used GUSi to determine the roles of gut microbial GUS enzymes in the colitis-promoting effects of TCS. We treated mice with vehicle or TCS, with or without co-administration of 1 mg/kg GUSi via oral gavage, and examined the development of dextran sodium sulfate (DSS)-induced colitis in mice (see the scheme of the experiment in Fig. 5a). We found that TCS exposure increased the severity of DSS-induced colitis in mice, akin to that reported previously[12]; however, this effect was abolished by co-administration of GUSi. Without GUSi, exposure to TCS exacerbated DSS-induced colitis: compared with vehicle, TCS treatment reduced colon length (Fig. 5b), caused more severe crypt damage (Fig. 5c), enhanced colonic infiltration of immune

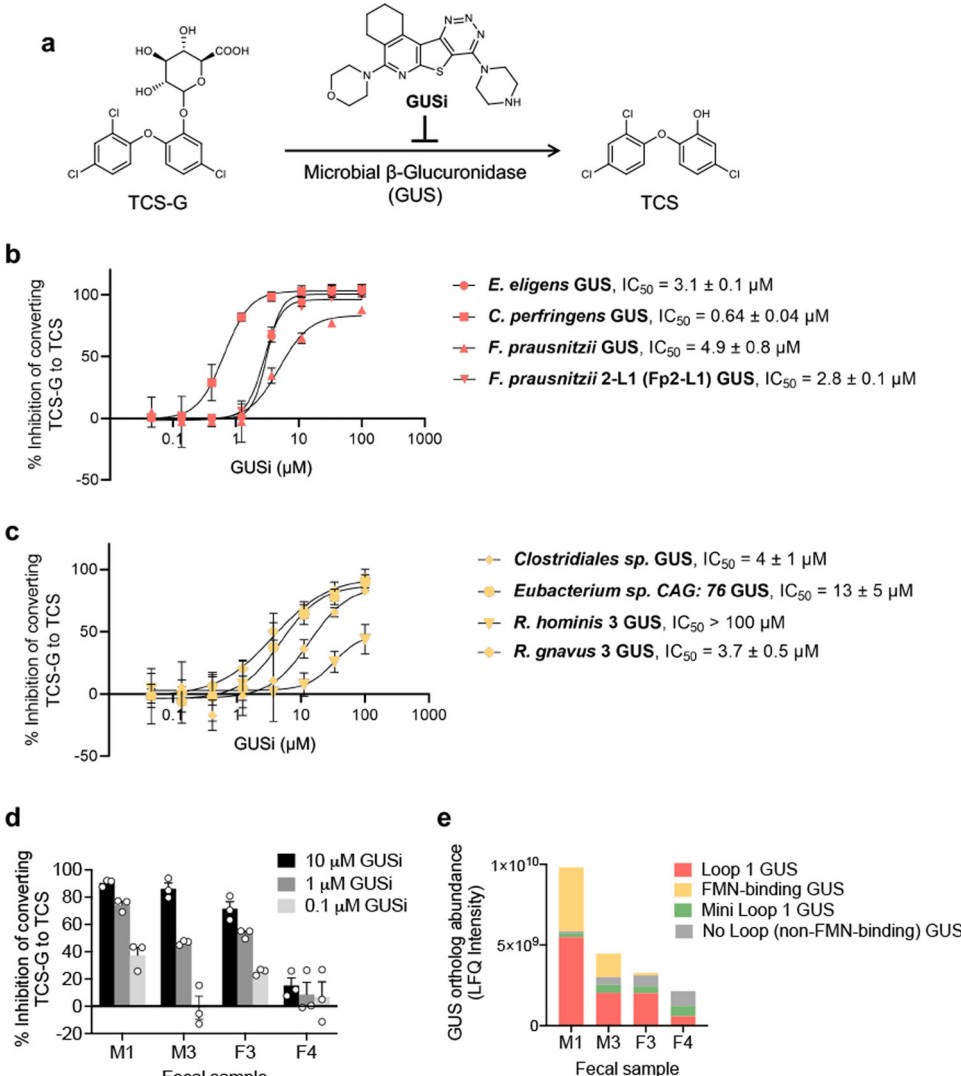

**Fig. 4 GUSi inhibits the conversion of TCS-G to TCS by gut microbial GUS enzymes. a** The effect of GUS inhibitor (GUSi; UNC10201652), on the conversion of TCS-G to TCS by gut microbial GUS enzymes. **b** GUSi inhibited the conversion of TCS-G to TCS catalyzed by purified Loop 1 GUS enzymes in vitro. **c** GUSi inhibited the conversion of TCS-G to TCS catalyzed by purified FMN-binding GUS enzymes in vitro. **d** GUSi inhibited the conversion of TCS-G to TCS catalyzed by enzymes extracted from human fecal samples ex vivo. **e** Abundance levels of GUS orthologs in human fecal samples as determined by activity-based probe-enabled proteomics. The data are mean ± SEM, n = 3 biological replicates. Source data are provided with this paper. TCS triclosan, TCS-G triclosan-glucuronide, GUS β-glucuronidase, GUSi β-glucuronidase inhibitor.

cells, including CD45[+] leukocytes, CD45[+] F4/80[+] macrophages, and CD45[+] Gr1[+] neutrophils (Fig. 5d), and increased expression of pro-inflammatory genes (*Tnf-a, Mcp-1, Il-6, Il-17,* and *Il-23*) in the colon (Fig. 5e). However, with co-administration of GUSi, the colitis-enhancing effects of TCS were eliminated across all measures (Fig. 5b–e). Thus, the inhibition of gut microbial GUS enzymes abolishes the colitis-enhancing effects of TCS, supporting the conclusion that GUS enzymes produced by the intestinal bacteria are required for the gut toxicity of TCS.

To validate GUSi-mediated target engagement, we first tested whether GUSi can reach the gut. LC-MS/MS studies showed that 2 days after the final oral administration of GUSi, the GUSi compound was detected in the colon tissues of the treated mice (Supplementary Fig. S15a, b). Next, we analyzed the concentrations of TCS (the product of GUS) and TCS-G (the substrate of GUS) in gut tissues. LC-MS/MS examination showed that GUSi treatment significantly reduced the ratio of TCS to TCS-G in colon digesta of the treated mice (Supplementary Fig. S15c). Thus, orally administered GUSi reaches the gut and suppresses

gut microbial GUS-mediated conversion of TCS-G to TCS, supporting the target engagement of GUSi in mice and helping to validate the conclusion that gut microbial GUS enzymes are required for the gut toxicity of TCS.

The results above suggest that gut microbial GUS-catalyzed conversion of TCS-G to TCS drives colitis, implicating that TCS, but not TCS-G, induces colonic inflammation. To test this, we studied the effects of TCS vs. TCS-G on inducing colonic inflammation in vitro. Intestinal epithelial (MC38) cells were treated with 1 μM of TCS or TCS-G and inflammatory responses were studied. The 1 μM concentration was chosen based on data presented above showing that TCS in the stool of TCS-exposed humans was up to ~1000 pmol/g (~1 μM, see Supplementary Fig. S2a and Supplementary Table S1). We find that treatment with TCS, but not TCS-G, increased gene expression and medium concentration of the pro-inflammatory cytokine IL-6 in MC38 cells (Supplementary Fig. S16). These data support the conclusion that TCS exerts direct pro-inflammatory effects on cultured intestinal epithelial cells, while the glucuronidated form of TCS

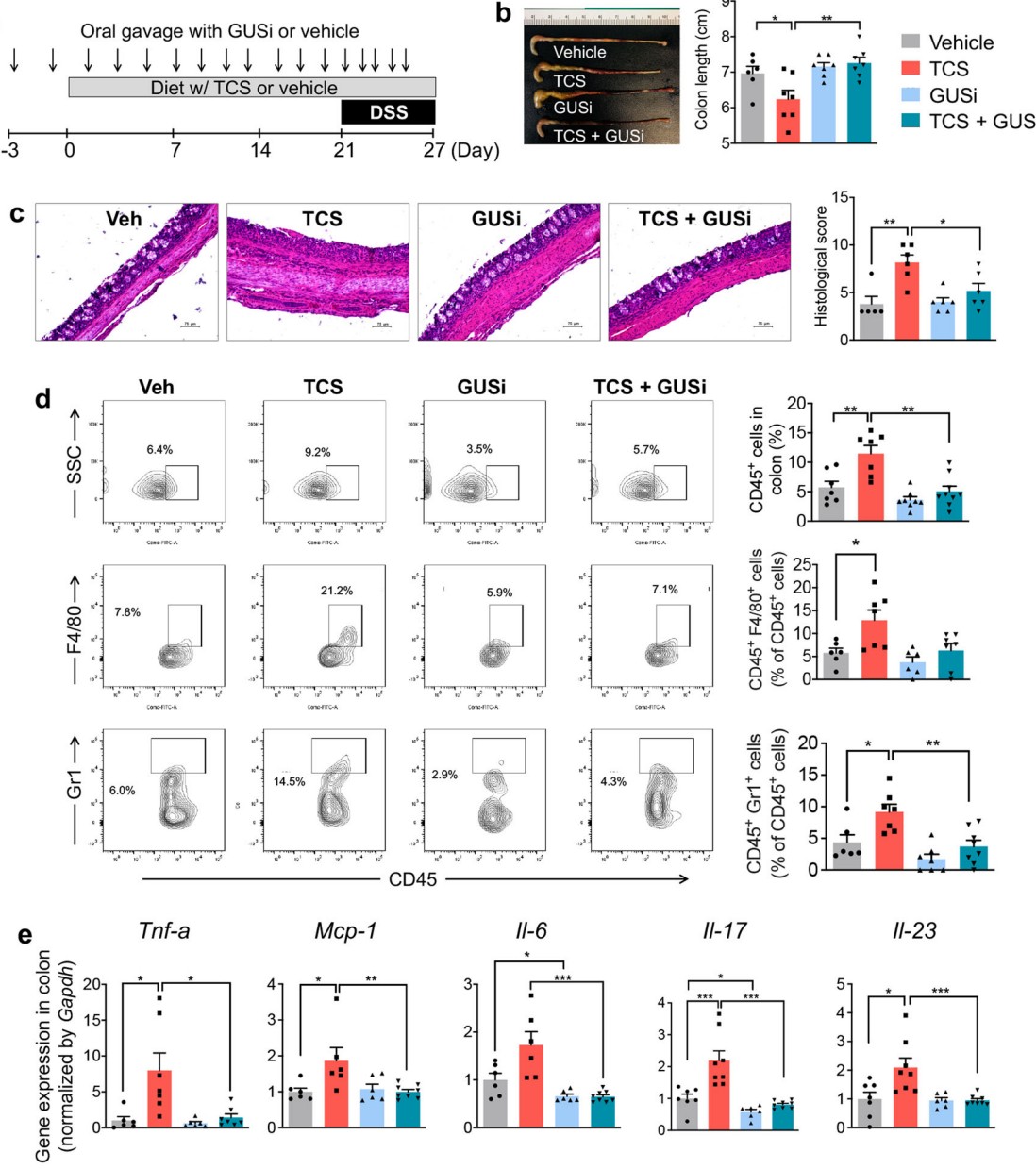

**Fig. 5 Inhibition of gut microbial GUS enzymes abolishes the colitis-promoting effects of TCS. a** C57BL/6 mice were treated with 80 ppm TCS or vehicle via diet, with or without co-administration of GUSi via oral gavage, then stimulated with DSS to induce colitis. **b** GUSi protects against the colon shortening effects of TCS ($n = 6$–8 mice per group). **c** GUSi reduces the crypt damaging effects of TCS. Left: representative H&E histological images of colon (scale bar = 75 μm). Right: quantification of histology score ($n = 5$–6 mice per group). **d** GUSi protects against immune cell infiltration induced by TCS. Left: representative FACS contour plots. Right: quantification of immune cells in the colon ($n = 6$–8 mice per group). **e** qRT-PCR analysis shows GUSi's reduction of TCS-induced pro-inflammatory gene expression in the colon ($n = 6$–8 mice per group). The data are mean ± SEM. To compare two groups, Shapiro–Wilk test was used to verify the normality of data; when data were normally distributed, statistical significance was determined using two-sided $t$ test; otherwise, significance was determined by Wilcoxon–Mann–Whitney test. *$P < 0.05$, **$P < 0.01$, ***$P < 0.001$. Source data are provided with this paper. TCS triclosan, GUSi β-glucuronidase inhibitor.

does not. Overall, the results from animal and cell culture studies support the conclusion the conversion of TCS-G to TCS by gut microbial GUS enzymes contributes to the pro-inflammatory effects of TCS in the mammalian intestine.

## Discussion

Proposed in 2013 and finalized in 2016, the U.S. FDA banned the marketing of TCS in over-the-counter antiseptic products intended to be used with water, including soaps. The concerns

cited focused on bacterial resistance and hormonal effects[10,11]. However, this restriction did not extend to toothpaste and other products capable of reaching the human gastrointestinal tract[10,11]. Our recent study showed that exposure to TCS exacerbates colitis in mouse models through gut microbiota-dependent mechanisms[12]. Here, we elucidate the molecular mechanisms by which gut microbiota contributes to the metabolic activation and subsequent gut toxicity of TCS. Our central finding is that specific gut microbial enzymes, notably, gut microbial GUS proteins of the Loop 1 and FMN-binding clades,

mediate the colonic reactivation of TCS from its inactive glucuronide metabolite, and in doing so they drive the gut toxicity of TCS. We identified the microbial enzymes involved using in vitro and ex vivo proteomic studies, and we define the molecular motifs required to metabolically activate TCS using crystal structures. Finally, we show that targeted inhibition of gut microbial GUS enzymes abolishes the colitis-promoting effects of TCS in mice, establishing the essential roles of specific microbial proteins in TCS toxicity. Together, these results define an axis of transformation previously unknown for prevalent environmental compounds like TCS.

Previous research regarding the metabolism of TCS, as well as many other environmental compounds, has focused on the metabolic processes in mammalian host tissues (e.g., liver), while their metabolic fates in the gut tissues are not well characterized[10,14]. Here, we showed that after TCS exposure in mice, the dominant compound in most host tissues is its conjugated metabolites such as TCS-G, akin to that reported previously[10,14]; however, the dominant compound in gut is free TCS. We treated mice with varying doses of TCS (1, 10, and 80 ppm TCS in diet) and found that at all tested doses, the gut tissues had similar metabolic profiles of TCS and were dominated by free TCS. In addition, we found that after the mice were exposed to TCS, notably at the lower doses (1 and 10 ppm in diet), the concentrations of TCS in mouse gut tissues are comparable or within several folds of the concentrations of TCS observed in the stool of TCS-exposed human subjects (see mouse data in Fig. 1b and human data in Supplementary Table S1). This result supports that it is feasible to use animal experiments to model human exposure to TCS, though we acknowledge that there are many challenges to use mouse models to study human exposure to consumer chemicals such as TCS. In addition, we found that after TCS exposure in humans, the human stool samples also exhibited the same TCS metabolic profile as we observed in the animal experiments and contained a high abundance of free TCS. Taken together, these results support that compared with other organs, the gut tissue has a unique profile of TCS metabolism. Using a combination of approaches including in vitro culturing of gut bacteria, antibiotic-mediated suppression of gut bacteria in vivo, and germ-free mice, we found that gut microbiota converts TCS-G to TCS in the colon and therefore contribute to the unique metabolic profile of TCS in the colon. Overall, these results support a model that after TCS exposure, it is metabolized in host tissues (notably the liver) and is converted to the conjugated metabolites such as TCS-G, which are then released to the intestines and are subjected to bacterial de-glucuronidation in the colon[38]. Other gastrointestinal factors, such as intestinal mobility and food intake, have been shown to modulate drug pharmacokinetics[40,41], and these factors could also affect the metabolic fates of TCS in the gut. Besides TCS, other environmental compounds could also have a distinct metabolic profile in gut tissues due to the metabolic activities of gut microbes, highlighting the importance of incorporating the microbiota in our understanding of environmental toxicology.

To date, the specific gut microbial enzymes involved in the toxicity of environmental pollutants remain largely unknown[42]. This is partially explained by the diversity of the gut microbial enzymes: the sequencing data from the Human Microbiome Project suggests that the human and mouse gut microbiotas contain hundreds of unique gut microbial GUS enzymes, which have different substrate specificities varying from small compounds to macromolecules[27,32]. Novel gut microbial GUS enzymes could be identified from further microbiota sequencing and/or functional characterization. Such variation in substrate specificity among GUS enzymes is due in part to the length and positioning of loops enclosing the GUS active site[25,28,31]. Using

sequence data from the Human Microbiota Project database, we binned GUS enzymes into seven structural classes based on their loop architecture[27]. We created a panel of 32 purified gut microbial GUS enzymes, which represent the seven classes, for in vitro enzymatic assays[29,30,34]. Using this strategy, we observed that Loop 1 and FMN-binding GUS enzymes were particularly efficient at processing TCS-G in vitro. This result suggests that these two classes were likely responsible for the majority of in vivo turnover of TCS-G as well. In support of this notion, using the approach of activity-based probe-enabled proteomics, we found that Loop 1 GUS, but not other classes such as Mini loop 1, No loop (non-FMN binding), and No Loop GUS, is correlated with TCS-G turnover in fimo. We were surprised, however, that there was no correlation between FMN-binding GUS enzymes and TCS-G turnover in fimo. One potential explanation could be that Loop 1 GUS enzymes vary mainly in the contiguous loop 1 sequence motif, which is only 15–20 residues in length. In contrast, FMN-binding GUS enzymes vary mainly in their large C-terminal domains of ~150 residues in length. To date, no structure of an FMN-binding GUS C-terminal domain has been reported, as they have remained mobile and unresolved in the structures determined thus far. Sequence identity does not appear to be sufficient to distinguish the differences between fast and slow-processing FMN-binding enzymes. For example, the sequence identity between the two fastest FMN-binding processors, Rh3 and *R. gnavus* 3 GUS, is 52.1%, while the sequence identity between the fastest and the slowest FMN-binding enzymes, Rh3 and Rh2 GUS, is 50.9%. It is possible that the abundance of efficient or fast FMN-binding GUS enzymes would correlate with in fimo TCS-G processing rates; but to date, because of the size of these C-terminal domains of FMN-binding GUS enzymes and our lack of structural knowledge about these domains, the specific motif(s) critical for TCS-G processing remain undefined. Overall, these results support that specific microbial GUS enzymes process TCS-G.

Through the discovery of the specific gut microbial enzymes involved in the metabolism and toxicology of TCS, our research could help to better evaluate its toxic potentials and clarify its individual effects in different populations. Based on our findings, upon TCS exposure, human subjects with a different abundance of Loop-1 or specific FMN-binding GUS enzymes could have varied colonic metabolism of TCS, resulting in inter-individual variations in biological responses to TCS exposure. Our previous study has revealed that there is significant inter-individual variability in the abundance of Loop-1 GUS orthologs in human fecal microbiotas[27]. In the sampled 139 human subjects from the Human Microbiota Project, ~40% did not have Loop-1 GUS in the fecal microbiota, and there was a wide range of abundance levels of Loop-1 GUS in those who have the Loop-1 GUS orthologs[27]. Here, we also showed that the fecal bacteria from different human subjects have a different abundance of Loop-1 GUS, leading to varied capacities to convert TCS-G to TCS. While future studies are needed to determine whether individuals with specific microbial GUS activities are more susceptible to the adverse effects of TCS exposure, such studies could chart the metabolic individuality of TCS and clarify the potential toxic effects of TCS on human health. More importantly, these studies will help to establish gut microbial enzymes as potential predictive markers for environmental toxicology.

Our results suggest that gut microbial GUS enzymes play critical roles in the metabolic reactivation and gut toxicity of TCS. Because genetic tools that specifically target gut microbial GUS enzymes are sparse[30,38], we used a pharmacological approach and employed GUSi as a chemical probe to elucidate the molecular mechanisms of TCS[24,37]. We showed that GUSi effectively inhibited GUS-mediated TCS-G processing in vitro and ex vivo,

with little effect on commensal microbes, growth of mammalian intestinal cells, or activity of mammalian GUS enzyme. These findings support that GUSi is highly selective toward the gut microbial GUS enzymes and therefore it is feasible to use GUSi to study the functional roles of gut microbial GUS enzymes in the gut toxicity of TCS. Next, we found that TCS exposure increased the severity of DSS-induced colitis in mice, however, the colitis-enhancing effects of TCS were abolished by co-administration of GUSi, confirming that the gut microbial GUS is required for the gut toxicity of TCS. Besides the DSS-induced colitis model, our previous study showed that TCS exposure exacerbated piroxicam-induced colitis in specific pathogen-free (SPF) $Il\text{-}10^{-/-}$ mice[12]. The conventionally housed $Il\text{-}10^{-/-}$ mice develop spontaneous colitis, and this spontaneous model can better model human IBD compared with the piroxicam-induced colitis model in SPF $Il\text{-}10^{-/-}$ mice[43]. It would be important to determine whether TCS exposure exacerbates colitis in the spontaneous $Il\text{-}10^{-/-}$ model and to elucidate the extent to which microbial GUS enzymes contribute to the biological effects of TCS in the spontaneous $Il\text{-}10^{-/-}$ model. Finally, we found that TCS (the product of GUS) exerts direct pro-inflammatory effects on cultured intestinal epithelial cells, while TCS-G (the substrate of GUS) was biologically inactive. Overall, these results support that GUS-mediated de-glucuronidation reaction leads to accumulation of free TCS in the colon and contributes to its pro-inflammatory effects in vivo. Since GUS-mediated de-glucuronidation is a common metabolic reaction involved in xenobiotic metabolism[38], and has been suggested to be the fourth phase of drug and xenobiotic metabolism[44], our findings may also apply to other environmental chemicals.

In summary, here we connect specific gut microbial enzymes with the metabolic reactivation of TCS in the colon and show that these enzymes drive adverse events caused by TCS. The data presented will help to better evaluate the individual effects of TCS in different populations. They also suggest that the safety of TCS and related compounds should be reconsidered given their potential for intestinal damage. Beyond TCS, it seems likely that gut microbial enzymes could contribute to the metabolism and toxicology of other chemicals, highlighting the critical importance of incorporating the microbiota in our understanding of environmental toxicology and mechanisms of disease.

## Methods

**Ethical statement.** The animal experiments were conducted in accordance with protocols approved by the Institutional Animal Care and Use Committee of the University of Massachusetts (Amherst, MA) and Massachusetts Host-Microbiome Center at the Brigham and Women's Hospital (Boston, MA). The analysis of the de-identified human urine and stool samples, which are from a previous human study (ClinicalTrials.gov identifier NCT01509976)[15], were conducted in accordance with the protocol approved by the Institutional Review Board of Stanford University. All subjects provided informed consent for their specimens to be used for studies of the microbiome; samples had been deidentified by the time this specific project was undertaken.

**Chemicals.** Triclosan (TCS, 99% purity) was purchased from Alfa Aesar (Haverhill, MA). TCS-glucuronide (TCS-G, 95% purity) and TCS-sulfate (95% purity) were from Santa Cruz Biotechnology (Dallas, TX). Stable isotope-labeled triclosan ($^{13}C_{12}$-TCS, 99% purity) was obtained from Cambridge Isotope Laboratories (Andover, MA).

### Animal experiments

*Animal experiment 1: LC-MS/MS profiling of TCS metabolism in mice.* C57BL/6 male mice (age = 6 weeks) were purchased from Charles River and maintained in a specific pathogen-free animal facility. The mice were treated with a modified AIN-93G diet which contains 1, 10, or 80 ppm TCS for 4 weeks, then the mice were sacrificed to harvest tissues for LC-MS/MS analysis. The composition of the diet is casein (200 g/kg), L-cystine (3 g/kg), sucrose (100 g/kg), dyetrose (132 g/kg), cornstarch (397.486 g/kg), cellulose (50 g/kg), mineral mix #210025 (35 g/kg), vitamin mix #310025 (10 g/kg), choline bitartrate (2.5 g/kg), corn oil (70 g/kg), and vitamin A palmitate (0.016 g/kg)[12]. The ingredients for the preparation of the diet,

except corn oil, were purchased from Dyets Inc. (Bethlehem, PA). The commercial sample of corn oil (Mazola, ACH Food company) was purchased from a local market in Amherst, MA, and purified by a silicic acid-activated charcoal chromatography to remove any pre-existing lipid oxidation compounds, then the purified oil was fortified with 400 ppm tocopherols, flushed with $N_2$, and stored at −80 °C until use.

*Animal experiment 2: effects of antibiotic suppression of gut microbiota on TCS colonic metabolism in mice.* C57BL/6 male mice (age = 6 weeks) were given drinking water with or without an antibiotic cocktail (a mixture of 1.0 g/L ampicillin and 0.5 g/L neomycin) throughout the entire experiment. This antibiotic composition was used in previous studies by others[20,21] and us[22,23]. After 5 days, the mice were treated with a modified AIN-93G diet which contains 80 ppm TCS (see diet composition in animal experiment 1 above). After another 4 weeks, the mice were sacrificed, and their tissues were collected for LC-MS/MS analysis.

*Animal experiment 3: effects of antibiotic suppression of gut microbiota on the kinetics of TCS colonic metabolism in mice.* C57BL/6 male mice (age = 6 weeks) were supplied with drinking water with or without the antibiotic cocktail for 7 days, then the mice were treated with a one-time oral gavage of 8 mg/kg TCS which was dissolved in polyethylene glycol 400 (PEG-400). At t = 4, 8, 12, and 24 h post the oral gavage, the mice were sacrificed to harvest tissues for analysis. The Area under the curve (AUC) (Fig. 2) was calculated using GraphPad Prism software, Version 9.1.2 (225) (https://www.graphpad.com/scientific-software/prism/) with the parameters as follows: the baseline is set as $Y = 0$ and the peaks that are less than 10% of the distance from minimum to maximum Y are ignored.

*Animal experiment 4: comparison of TCS colonic metabolism in conventional mice vs. germ-free mice.* Conventional or germ-free male mice, established on C57BL/6 or Swiss Webster background, were treated with a one-time oral gavage of 8 mg/kg TCS which was dissolved in PEG-400. At t = 4–8 h post the oral gavage, the mice were sacrificed to harvest tissues for LC-MS/MS analysis.

*Animal experiment 5: effects of GUSi on colitis-enhancing effects of TCS in mice.* C57BL/6 male mice were orally gavaged with a specific GUS inhibitor (GUSi) UNC10201652 (dose = 1 mg/kg) or vehicle (a mixed solvent of 1:9 DMSO and saline) every other day throughout the experiment, as described previously[29,30]. After 3 days, the mice were treated with a modified AIN-93G diet which contains 80 ppm TCS or vehicle (PEG-400) until the end of the experiment. After another 3 weeks, the mice were stimulated with 2% DSS (molecular weight = 36–50 KDa, MP Biomedicals, Solon, OH) in drinking water for 6 days to induce colitis. At end of the experiment, the mice were sacrificed for analysis.

*Animal experiment 6: effects of GUSi on gut inflammation and gut microbiota in mice.* C57BL/6 male mice were orally gavaged with GUSi UNC10201652 (dose = 1 mg/kg) or vehicle (a mixed solvent of 1:9 DMSO and saline) every other day for 24 days (the same treatment scheme as in animal experiment 5). At end of the experiment, the mouse feces were collected and subjected to sequencing, and the mice were sacrificed for biochemical analysis.

**Detection of TCS and its metabolites by LC-MS/MS.** Mouse tissues and human stool samples were placed in homogenizer tubes with beads and 1 mL methanol, then homogenized using a bead-disruptor (OMNI International, Kennesaw, GA). Samples were centrifuged at 10,000 rpm for 3 min. The supernatant was collected and then centrifuged again at 14,000 rpm for 5 min. In total, 500 μL of the supernatant was then collected, and vacuum centrifuged to dryness. For bacterial broth (50 μL) and human urine (100 μL), each sample was combined with 1 mL methanol and placed on ice. After 10 min on ice, samples were centrifuged at 14,000 rpm for 5 min. 500 μL of the supernatant was then collected and vacuum centrifuged to dryness. Stable isotope-labeled $^{13}C_{12}$-TCS was used as the surrogate standard during the extraction. The extracts were re-dissolved in methanol with the amount that was proportional to sample weights or volumes, then centrifugated (14,000 rpm, 15 min, 4 °C) before the LC-MS/MS analysis.

TCS, TCS-G, and TCS-sulfate in the samples were quantified using a Thermo Scientific Dionex Ultimate 3000 ultrahigh performance liquid chromatography (UHPLC) system coupled with a TSQ Quantiva Triple Quadrupole Mass Spectrometer. ACQUITY UPLC C18 column (1.7-μm particles, 2.1 × 100 mm, Waters) was used for chromatographic separation. Data acquisition was performed by multiple reaction monitoring (MRM) in negative ionization mode. Details of the instrumental methods are provided in Supplementary Table S4. The data were analyzed using Xcalibur software (version 4.1, Thermo Fisher Scientific).

The spike recoveries of the three target compounds in the matrixes of mouse colon digesta were determined. The recoveries (%, mean ± SEM) were 101.6 ± 8.9 and 95.6 ± 3.4 for TCS, 91.6 ± 5.4 and 87.1 ± 5.9 for TCS-G, 95.1 ± 1.4 and 96.9 ± 6.0 for TCS-sulfate, based on two spiked levels of 2 pmol/mg and 10 pmol/mg, respectively ($n = 3$ replicates). No significant differences were found among these three compounds. Therefore, $^{13}C_{12}$-TCS was used for the signal correction of TCS, TCS-G, and TCS-sulfate, and it is a strategy for the absolute quantitation of analytes when internal standards are unavailable[12,45]. For the quantification of

TCS, TCS-G, and TCS-sulfate by LC-MS/MS in the different experiments, blank samples from the control group without TCS exposure were used as the matrixes for calibration curve standards. During the instrumental analysis, the matrix calibration curve was performed at the beginning and at the end of every sample batch. All reported concentrations were determined based on a standard curve with 7–10 data points.

**Isolation of bacteria from mouse tissues and human stool samples.** Mouse fecal tissues and human stool samples were collected, dissolved in sterile PBS with 0.05% L-cysteine, then centrifuged at $900 \times g$ for 5 min. The supernatant containing culturable bacteria was then fermented at 37 °C in MRS broth in an anaerobic cabinet (Whitley A35 anaerobic workstation, Don Whitley Scientific) under an atmosphere of 85% $N_2$, 10% $CO_2$, and 5% $H_2$. In addition, the remaining supernatant (~0.6 mL) containing culturable bacteria was then mixed with sterile 50% glycerol (0.3 mL) and stored at −80 °C as stock for future experiments.

**Protein gene synthesis, expression, and purification.** All genes were codon-optimized for *E. coli* expression, synthesized, and ligated into a pLIC-His vector by BioBasic. Genes were transformed into BL21-G *E. coli* competent cells. A 100 mL culture was grown overnight at 37 °C in LB broth with ampicillin (100 μg/mL) and shaking at 215 RPM. The following day, 50 mL overnight culture was added to 1.5 L of LB broth with ampicillin and ~40 μL Antifoam 204. For FMN-binding enzymes, 500 μM FMN was added to the culture flask. The culture was incubated at 37 °C and 215 RPM until it reached an OD of 0.6. The culture was then induced with 1-thio-β-D-galactopyranoside (100 μM) and incubated overnight at 18 °C.

Cells were pelleted at $4,500 \times g$ for 20 min at 4 °C in a Sorvall (model RC-3B) swinging bucket centrifuge. Pellets were resuspended in 35 mL Buffer A (20 mM potassium phosphate, 50 mM imidazole, 500 mM NaCl, pH 7.4. For FMN-binding enzymes, buffer contained 50 μM FMN) with DNAse, lysozyme, and one EDTA-free protease inhibitor tablet (Roche). The resuspension was sonicated twice using 1 s pulses for 1.5 min and the resultant suspension was pelleted at $17,000 \times g$ for 45 min in a Beckman Coulter J2-HC centrifuge. The supernatant was syringe-filtered using a 0.22-μm filter.

The filtrate was flowed over a 5-mL nickel-nitrilotriacetic acid HP column (GE Healthcare) using the Aktaexpress FPLC (Amersham Bioscience) and washed with Buffer A. Protein was eluted using a linear gradient of Buffer A to Buffer B (20 mM potassium phosphate, 500 mM imidazole, 500 mM NaCl, pH 7.4. For FMN-binding enzymes, buffer contained 50 μM FMN). Fractions containing the protein of interest were collected and applied to a HiLoad 16/60 Superdex 200 gel-filtration column (GE Life Sciences). Samples were eluted in S200 buffer (20 mM HEPES, 50 mM NaCl, pH 7.4). Fractions containing the protein of interest were analyzed via SDS-PAGE. Those with >95% purity were combined and concentrated to ~10 mg/mL using 50 kDa cutoff molecular weight centrifuge concentrators (EMD Millipore). Samples were snap-frozen using liquid nitrogen and stored at −80 °C.

**Site-directed mutagenesis.** All mutants were created using site-directed mutagenesis. Primers were synthesized by IDT Technologies (Supplementary Table S5). Mutant plasmids were sequenced by Eton Biosciences to confirm mutation incorporation. Mutant proteins were purified using the same purification protocol described above.

**Fecal extract preparation, proteomics, and analysis.** Fecal extracts were prepared and proteomic analysis was performed exactly as previously described[35]. In total, 10 g of frozen human fecal sample was thawed and added to 25 mL extraction buffer (25 mM HEPES, 25 mM NaCl pH 6.5 with Roche Complete protease inhibitor tablet) and 0.5 g autoclaved garnet beads. The mixture was vortexed until homogenous and centrifuged at low speed ($300 \times g$ for 5 min at 4 °C). The supernatant was decanted. In all, 25 mL buffer was added to the pellet and the vortex and centrifugation steps repeated. The supernatants from these steps were then combined and centrifuged at low speed for two more cycles. The resultant supernatant was sonicated twice using 1 s pulses for 1.5 min and the lysate was centrifuged at $17,000 \times g$ for 20 min in a Beckman Coulter J2-HC centrifuge. The decanted lysate was then washed with several exchanges of extraction buffer to remove metabolites and small molecules. Total protein concentration was quantitated using a Bradford assay. The fecal lysate was diluted to 1 mg/mL concentration and snap-frozen in small aliquots.

In all, 3.5 mg purified fecal extract was incubated with 10 μM biotin-activity-based probe complex in 500 μL extraction buffer with 1% dimethyl sulfoxide for 1 h at 37 °C. To quench, 125 μl 10% sodium dodecyl sulfate (SDS) was added and samples were heated to 95 °C for 5 min. Samples were cooled on ice and washed with extraction buffer containing 0.05% SDS three times by centrifugation for 5 min at $14,000 \times g$ in 1.5 mL Amicon 10 K cutoff spin concentrators. After centrifugation, the total volume was brought to 1 mL using extraction buffer with 0.05% SDS. In total, 15 μL streptavidin sepharose beads (GE) were added and samples incubated at room temperature for 1 h. Beads were then washed 3 times with 300 μL extraction buffer with 0.1% SDS, three times with 300 μL extraction buffer alone, and three times with 300 μL 50 mM $NH_4HCO_3$. Samples were centrifuged at $400 \times g$ for 2 min at 4 °C between washes, and the supernatant

decanted. Beads were then resuspended in 100 μL 50 mM $NH_4HCO_3$ and stored at −20 °C.

The resultant bead mixture was added to 0.5% Rapigest (Waters) in 50 mM $NH_4HCO_3$ and reduced with dithiothreitol at 65 °C for 30 min. 2-chloroacetamide was then added and the mixture was incubated in the dark for 20 min at room temperature. Mixtures were centrifuged at $200 \times g$ for 2 min at room temperature to pellet beads. The supernatant was decanted and trypsinized with 2.5 μg of trypsin overnight at 37 °C. Mixtures were then concentrated to 100 μL in a speedvac and desalted with C18 desalting columns (Thermo Scientific). Samples were reconcentrated using the speedvac and 100 μL LC-Optima MS grade water was added to solubilize samples. Samples were extracted with ethyl acetate and concentrated in the speedvac. The Pierce QFP assay (Thermo) was used to quantify and normalize peptides.

Trypsinized peptides were separated using reverse-phase nano-high-performance liquid chromatography (nano-HPLC) coupled with a nanoACQUITY ultraperformance liquid chromatography (UPLC) system (Waters Corporation). Peptides were trapped and separated in a 2 cm column (Pepmap 100; 3-m particle size and 100-Å pore size), and a 25-cm EASYspray analytical column (75-m inside diameter [i.d.], 2.0-m C18 particle size, and 100-Å pore size) at 300 nL/min and 35 °C, respectively. A 60 min. gradient of 2% to 25% buffer B (0.1% formic acid in acetonitrile) was conducted on an Orbitrap Fusion Lumos mass spectrometer (Thermo Scientific) with ion source at 2.4 kV and ion transfer tube at 300 °C. MS scans from 350 to 2000 $m/z$ were acquired using the Orbitrap at a resolution of 120,000 and 1e6 AGC target. MS2 spectra were collected with 1.6 $m/z$ isolation width and were analyzed using the 3 s TopSpeed CHOPIN method by the Orbitrap or the linear ion trap depending on peak charge and intensity[46]. Orbitrap MS2 scans were acquired at 7500 resolution with a 5e4 AGC and 22 ms maximum injection time after HCD fragmentation with normalized energy of 30%. Rapid linear ion trap MS2 scans were obtained with a 4e3 AGC, 250 ms maximum injection time after CID 30 fragmentation. Precursor ions were chosen based on intensity thresholds (>1e3) from the full scan and on charge states with a 30-s dynamic exclusion window. Polysiloxane 371.10124 was used as the lock mass. The mass spectrometry proteomics data have been deposited to the ProteomeXchange Consortium via the PRIDE[47] partner repository with the dataset identifier PXD025887.

Data were processed using Metalab verson 1.1.1[48] with MaxQuant version 1.6.2.3[49] to identify peptides and protein groups. The integrated reference catalog of the human gut microbiome database[50] combined with the UniProtKB/Swiss-Prot human sequence database (downloaded Feb 1, 2017)[51] with total 9,920,788 sequences was used as the database search. Search parameters were static carbamidomethyl cysteine modification, specific trypsin digestion with up to two missed cleavages, variable protein N-terminal acetylation and methionine oxidation, match between runs, and label-free quantification (LFQ) with a minimum ratio count of 2. A false discovery rate (FDR) of 1% was used for filtering protein identifications, and potential contaminants and decoys were removed.

GUS enzymes in each database were identified by pairwise alignment to the representative EcGUS *Escherichia coli* (EcGUS, UniProt: P05804), *Clostridium perfringens* (CpGUS, UniProt: Q8VNV4), *Streptococcus agalactiae* (SaGUS, UniProt: Q8E0N2), and *Bacteroides fragilis* (BfGUS, PDB: 3CMG). A sequence identity threshold 28% was required with at least one of the four representative proteins. In addition, all conserved residues had to be present and correctly aligned to the representative protein that passed the identity threshold. The conserved residues were: EcGUS E413, E504, N566, K568; CpGUS E412, E505, N567, K569; SaGUS E408, E501, N563, K565; and BfGUS E395, E476, N547, K549. GUS loop classes were determined by multiple sequence alignment with representative proteins, followed by examination of each sequence for specific loop criteria as defined by Pollet et al.[27] and Pellock et al.[52].

**In vitro UDH assay.** TCS-G was resuspended in 100% DMSO to a concentration of 10 mM. The assay reaction mixture consisted of 10 μL NAD+ (2 mM final), 5 μL uronate dehydrogenase (1 μM final), 5 μL various GUSs (50 nM final), and 30 μL TCS-G (200 μM final). Components were previously diluted in assay buffer (50 mM HEPES, 50 mM NaCl, various pH) or (50 mM sodium acetate, 50 mM NaCl, various pH). The pH of each reaction was determined using the optimal pH of the reaction as determined using pNPG[53]. Reactions were incubated at 37 °C for 30 min, and absorbance was monitored continuously at 340 nm using a BMG Labtech PHERAstar plate reader. The initial velocity of the reaction was fit using linear regression in MATLAB. Rates are the average of three biological replicates ±SEM.

**Catalytic efficiency assay.** Assay mixtures contained 10 μL GUS (various final concentrations, between 10–50 nM), 30 μL TCS-G (final concentrations between 30–120 μM), and 10 μL assay buffer (50 mM HEPES, 50 mM NaCl, various pH) or (50 mM sodium acetate, 50 mM NaCl, various pH). Control reactions replaced GUS with buffer. Reactions were quenched at five time points with 50 μL 25% trichloroacetic acid. Samples were centrifuged for 10 min at $16,000 \times g$, and the supernatant was subjected to analysis by HPLC on an Agilent 1260 Infinity II system using an Agilent InfinityLab Poroshell 120 C18 column (4.6 × 100 mm, 0.7 μm particle size). The column temperature was set to 38 °C with a flow rate of 0.9 ml/min and injection volume of 40 μL. Conditions were set to flow 98% A

(water with 0.1% formic acid) and 2% B (acetonitrile with 0.1% formic acid) for two minutes. A linear gradient was then set to flow to 98% B over 10 min and held for 4 min. Conditions were then ramped down to 98% A for 1 min and re-equilibrated at 98% A for 2 min. Analytes were detected using an Agilent DAD detector at a wavelength of 280 nm. Concentrations of TCS-G were determined using a standard curve of TCS-G (0-250 μM). Reaction curves were fit using linear regression, and the resultant initial velocities were plotted against substrate concentration to determine $k_{cat}/K_M$. Reported catalytic efficiencies are the average of three biological replicates ±SEM.

**In vitro IC$_{50}$ assay**. Reaction mixtures containing 10 μL GUS (10 nM final), 10 μL TCS-G (200 μM final), 5 μL inhibitor (various concentrations), and 25 μL buffer (50 mM HEPES, 50 mM NaCl, various pH) or (50 mM sodium acetate, 50 mM NaCl, various pH) were incubated for 10 min and quenched with 50 μL 25% trichloroacetic acid. Samples were centrifuged 10 min at 16,000 × g, and the supernatant was analyzed using the method described for the catalytic efficiency assay. Inhibition was calculated by the equation below:

$$\% \text{ inhibition} = 100 \times (1 - (AUC_{inh} - AUC_{max})/(AUC_{min} - AUC_{max})) \quad (1)$$

where AUC$_{min}$ is the signal of the uninhibited reaction, AUC$_{max}$ is the signal of the 100% inhibited reaction, and AUC$_{inh}$ is the signal of the reaction at a given concentration of inhibitor. Percent inhibition values were plotted against the log of inhibitor concentration, and GraphPad Prism 8.0 was used to determine IC$_{50}$ values.

**In fimo assay**. Reaction mixtures contained 5 μL fecal extract (0.1 mg/mL final), 30 μL TCS-G (200 μM final), and 15 μL assay buffer (25 mM HEPES, 25 mM NaCl, pH 6.5). Reactions were quenched at five time points with 50 μL 25% trichloroacetic acid. Samples were centrifuged 20 min at 16,000 × g in a tabletop centrifuge, and the supernatant was analyzed via the same HPLC method described for the catalytic efficiency assay. Reaction rates were determined by fitting progress curves using linear regression and are expressed as initial turnover rates (μM/s). Controls contained fecal extract that had been heat-killed at 95 °C for 5 min. Reported rates are the average of three biological replicates ±SEM.

**In fimo inhibition assays**. Reaction mixtures contained 5 μL fecal extract (0.1 mg/mL final), 10 μL TCS-G (200 μM final), 5 μL GUSi (10, 1, or 0.1 μM final), and 30 μL assay buffer (25 mM HEPES, 25 mM NaCl, pH 6.5). Reactions were quenched at 30 min with 50 μL 25% trichloroacetic acid. Samples were centrifuged 20 min at 16,000 × g in a tabletop centrifuge, and the supernatant was analyzed via the same HPLC method described for the catalytic efficiency assay. Controls contained 5 μL buffer in place of GUSi. Inhibition was calculated using Eq. 1.

**Crystallography**. Crystals were produced using the sitting drop vapor diffusion method at 20 °C. Trays were set up using the Art Robbins Instruments Crystal Phoenix robot or an Oryx4 robot (Douglas Instruments) and Hampton Research three-well midi crystallization plates (Swissci). For *Roseburia hominis* 3 GUS, crystals were produced in a condition containing 100 nL 12.5 mg/mL Rh3 GUS and 200 nL 0.2 M LiCl, 20% PEG 3350. For *Faecalibacterium prausnizii* L2-1 GUS, FpL2-1 GUS at 15 mg/mL was preincubated with UNC10201652 (GUSi) and PNPG in tenfold excess prior to addition to the crystalline solution. Crystals formed in a condition containing 200 nL GUS and 100 nL 0.2 M potassium thiocyanate (KSCN), 20%(w/v) PEG 3350.

Crystals were cryo-protected using the crystal solutions as described above with 20% glycerol. Diffraction data were collected at 100 K at APS beamline 23-ID-D. Data were processed using XDS and structures were solved using molecular replacement in Phenix. For Rh3 GUS, the *R. hominis* 2 structure (PDB 6MVH) was used as a search model. For FpL2-1 GUS, a FpL2-1 model produced using the Phyre2 server was used as a search model[54]. Maps and models output from molecular replacement were run through the Autobuild function of Phenix (version 1.17.1-3660)[55]. Structures were refined using phenix.refine, and Coot (version 0.9.4) was used for manual, visual inspection, and ligand fitting[56]. Final PDB coordinates were deposited to the RCSB Protein Data Bank under the codes 7KGZ (Rh3) and 7KGY (FpL2-1 with GUSi).

**Docking of TCS-G using Schrodinger**. TCS-G docking into various GUS enzymes was carried out using the Schrödinger (Release 2020-1, http://www.schrodinger.com) induced fit docking pipeline. The Schrödinger Protein Preparation module was used to prepare proteins for docking by adding hydrogens, deleting water molecules more than 3 Å from the ligand, generating protonation states based on the protein's ideal pH (as determined previously via in vitro assays), and creating metal and disulfide bonds. Default settings were used with the exception of the pH protonation states. The wizard was used to preprocess the structure, remove waters, optimize H-bonds, and minimize the structure.

Ligands were prepared using the Ligprep module. Ionization states were generated at pH 6.5 ± 0.5. Induced fit docking was used to dock TCS-G into the active sites of each GUS enzyme. Using a previously solved structure of *E. eligens* GUS with glucuronic acid bound (PDB: 6BJQ), glucuronic acid was added into the

active site of each GUS using PyMOL by aligning each GUS to 6BJQ. The box center was then chosen as the glucuronic acid location with a box size of 30 Å. Core constraints were added to restrict docking of TCS-G to the existing glucuronic acid structure using the maximum common substructure. Glide redocking was performed at XP precision. Top docking poses were chosen based on the IFD Docking score and the Glide Score, as well as visual examination to confirm probable binding mode.

**Circular dichroism**. The protein stabilities of Rh3 GUS and Fp2-L1 GUS and their mutants were determined using the circular dichroism method[57]. Enzyme (0.125 mg/mL) in CD buffer (10 mM potassium phosphate pH 7.4, 100 mM potassium fluoride) was loaded into a 1-mm cuvette. The Chirascan Plus instrument (Applied Photophysis Limited) was used to acquire 1) scan spectra from 185 to 260 nM at 20 °C and 2) a melting profile at 193 nM from 20 to 94 °C. Spectra acquired with buffer alone were used to correct for background signal.

**Cell culture and treatment with TCS and TCS-G**. MC38 intestinal epithelial cells were cultured in Dulbecco's Modified Eagle Medium (DMEM) supplemented with 10% fetal bovine serum (FBS) and incubated at 37 °C under an atmosphere of 5% CO$_2$. MC38 cells were seeded at 20% confluency and left to settle overnight. Cells were then treated with either 1 μM TCS, TCS-G, or vehicle (DMSO). After 48 h, cells were harvested for RT-qPCR analysis. Cell medium was collected for ELISA analysis using the CBA Mouse Inflammation Kit (BD Sciences) according to the manufacturer's instructions, and data were acquired using a BD LSR Fortessa flow cytometer (BD Biosciences) and analyzed using FlowJo software (Treestar).

**ELISA of inflammatory biomarkers in plasma**. Blood samples were harvested via cardiac puncture and collected in blood collection tubes (Covidien). The plasma fractions were prepared by centrifugation of the harvested blood at 3000 × g for 5 min at 4 °C. The concentrations of cytokines in plasma were determined using the CBA Mouse Inflammation Kit (BD Biosciences) as described above.

**Reverse-transcriptase-qPCR of inflammatory biomarkers**. Total RNA of colon tissues and MC38 cells were isolated using Trizol reagent (Ambion) according to the manufacturer's instruction. RNA was reverse transcribed into cDNA using the High Capacity cDNA Reverse Transcription kit (Applied Biosystem) according to the manufacturer's instructions. In all, 20 μL PCR reactions were prepared using the Maxima SYBR Green Master Mix (Thermo Fisher Scientific), and qPCR was carried out using a DNA Engine Opticon System (Bio-Rad Laboratories). Mouse-specific primer sequences (Thermo Fisher Scientific) used to detect inflammatory biomarkers are listed in Supplementary Table S6. *Gapdh* expression was used as an internal control.

**Flow cytometry**. Distal colon tissues were dissected, washed with cold PBS, and digested with Hank's balanced salt solution (HBSS, Lonza) supplemented with 1 mM dithiothreitol (DTT) and 5 mM EDTA at 4 °C (colon epidermal cells). The released cells were stained with FITC-conjugated anti-mouse CD45 (BioLegend, Clone: 30-F11), PerCP/Cy5.5-conjugated anti-mouse F4/80 (BioLegend, Clone: BM8), and PE/Cy7-conjugated anti-mouse Ly-6G/Ly-6C (GR-1) (BioLegend, Clone: RB6-8C5) with a 1:100 diluted solution. Cells were stained with Zombie Violet$^{TM}$ dye (Zombie Violet$^{TM}$ Fixable Viability Kit; BioLegend) according to the manufacturer's instructions to exclude dead cells. Gating and cell identification strategies are as follows: briefly, cell doublets and clumps were eliminated using FSC-A gating and debris was eliminated using FSC-A vs SSC-A. Dead cells were gated out using Zombie Violet$^{TM}$ dye. Flow cytometry data were acquired on a BD LSR Fortessa$^{TM}$cell analyzer (Becton Dickinson, Franklin Lakes, NJ) and analyzed using FlowJo software (FlowJo, LLC). Gating strategies used for the identification of major immune cell populations are shown in Supplementary Fig. S17.

**Histological staining**. The dissected colon tissues were fixed in 10% neutral buffered formalin (Thermo Fisher Scientific) for 48 h. After dehydration, the tissues were embedded in paraffin and sliced (5 mm) by Rotary Microtome (Thermo Fisher Scientific). The slices were dewaxed in serial xylene and rehydrated through ethanol solutions, stained with hematoxylin and eosin (Sigma-Aldrich), and images were obtained under 200× magnification (BZ-X700 microscope, Keyence, Itasca, IL). The histologic scores were evaluated by a blinded observer according to the following measures: crypt architecture, degree of inflammatory cell infiltration, muscle thickening, goblet cell depletion, and crypt abscess. The histologic damage score is the sum of each individual score.

**DNA extraction**. DNA was extracted from mouse fecal samples using QIAmp DNA Stool Mini Kit (Qiagen, Valencia, CA) following instructions from the manufacturer with an additional bead-beating step. The quantity of the extracted DNA was measured using a NanoDrop Spectrophotometer (Thermo Fisher Scientific), and the quality was verified using gel electrophoresis. The DNA was then subjected to further analysis.

**Real-time PCR analysis of *16S rRNA* gene**. DNA extracted from mouse fecal samples were subjected to qPCR analysis using a DNA Engine Opticon System (Bio-Rad Laboratories, Hercules, CA). In all, 20 μL PCR reactions were made using the Maxima SYBR_green Master Mix (Thermo Fisher Scientific), and DNA was normalized to 5 ng/μL per reaction. The *16S rRNA* primers are in Supplementary Table S6.

**16S rRNA sequencing and analysis**. DNA quality was monitored on 1% agarose gels. The V3–V4 hypervariable regions of the bacteria *16S rRNA* gene were amplified with primers 341F (5'-CCTAYGGGRBGCASCAG-3') and 806R (5'-GGACTACNNGGGTATCTAAT-3'). PCR products were detected on 2% agarose gels by electrophoresis and purified using the Qiagen Gel Extraction Kit (Qiagen, Germany). Sequencing libraries were generated using NEBNext Ultra DNA Library Pre Kit for Illumina, following the manufacturer's recommendations and index codes were added. The library quality was assessed using the Qubit 2.0 Fluorometer (Thermo Scientific) and Agilent Bioanalyzer 2100 system. The library was sequenced on an Illumina platform and 250 bp paired-end reads were generated.
   To analysis difference in abundance patterns among samples, beta diversity using weighted UniFrac distance followed by Principal Coordinate Analysis (PCoA) and non-metric multidimensional scaling (NMDS) analysis. The alpha diversity was considered as the richness of the samples, number of OTUs present per treatment. We calculated Simpson, Shannon, Chao1 index using QIIME2 version 2019.7.0[58] and Phytools package 0.7 v[59] in R (R Development Core Team, 2014). Statistical tests were performed in R. Kruskal-Wallis rank-sum test was used to compare alpha diversity indexes among treatments. The beta diversity was calculated using Phytools 0.7 v. The weighted UniFrac distance method was used to create the matrix followed by PCoA to compare similarity among treatments. All plots were obtained using ggplot2[60].

**Cell proliferation assay**. Mouse (MC38) or human (Caco2 and HCT-116) intestinal cells were grown in DMEM medium fortified with 10% FBS (EMD Millipore Corporation). The cells were seeded in 96-well plates, then treated with GUSi or vehicle (0.2% v/v DMSO) for 24 h. Cell viability was determined using an MTT assay.

**Data and statistical analyses**. Data are mean ± SEM. For the comparison between two groups, Shapiro–Wilk test was used to verify the normality of data; when data were normally distributed, statistical significance was determined using two-sided *t* test; otherwise, significance was determined by Wilcoxon–Mann–Whitney test. The statistical comparison of three groups was analyzed using one-way ANOVA by Tukey's multiple comparisons. The statistical analyses were performed using SAS (version 9.3) statistical software and GraphPad Prism (version 8.0 or 9.0). $P < 0.05$ was considered statistically significant.

**Reporting summary**. Further information on research design is available in the Nature Research Reporting Summary linked to this article.

## Data availability

All data generated or analyzed during this study are included in this published article (and its supplementary information files). The mass spectrometry proteomics data have been deposited to the ProteomeXchange Consortium via the PRIDE partner repository with the dataset identifier PXD025887. Final PDB coordinates were deposited to the RCSB Protein Data Bank under the codes 7KGZ (Rh3) and 7KGY (FpL2-1 with GUSi). The 16S rRNA sequencing data have been deposited to the Sequence Read Archive with BioProject identifier PRJNA781381. Source data are provided with this paper.

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

## Acknowledgements

We thank Prof. D. Joseph Jerry at the Department of Veterinary and Animal Sciences of the University of Massachusetts Amherst for recording histology images. This research is supported by interdisciplinary faculty research award from the University of Massachusetts Amherst, USDA NIFA 2019-67017-29248 and 2020-67017-30844, and USDA/Hatch MAS00556 (to G.Z.), NIH/NIGMS R01s GM135218, GM137286 (to M.R.R.), General Research Fund (12303319) of Hong Kong Research Grants Council (to Z.C.), NIH/NIEHS R21 ES023371 (to J.P.), NIH/NCCIH R01 AT010229 (to H.X.), and NIH T32GM008570 and NSF DGE-1650116 (to M.E.W.). We acknowledge the NIH grant P30DK034854 and the use of the Harvard Digestive Disease Center's (HDDC's) core services, resources, technology, and expertise.

## Author contributions

J.Z. and K.Z.S. performed and analyzed the cell culture and animal experiments. E.Z. and K.Z.S. performed and analyzed the fecal bacterial experiments. G.W. performed animal experiments. H.Z., Y.L., and J.Y. performed and analyzed LC-MS/MS experiments. V.Y. performed germ-free mouse experiments. K.C. and J.G.G. analyzed *16S rRNA* sequencing data. A.P. and D.K. performed the statistical analysis. J.P. and T.D.H. provided human samples and analyzed the data. L.M.M. analyzed FACS data. Z.L. analyzed the data and prepared the figures. H.X. and L.L. analyzed the data and helped in study design. M.E.W. and J.B.S. performed crystallography experiments. M.E.W. and V.V.B. performed kinetics assays in vitro and in fecal lysates, docked substrates into proteins using Schrodinger, and designed and created mutations using site-directed mutagenesis. V.B. and H.S.O. designed and provided the activity-based probe for proteomics experiments. P.B.J., E.W.C., M.B.M., and D.G. performed and analyzed proteomics experiments. J.Z., M.E.W., K.Z.S., H.Z., Z.C., M.R.R., and G.Z. designed the study and wrote the manuscript.

## Competing interests

M.R.R. is a Founder of Symberix, Inc., which is developing microbiome-targeted therapeutics. M.R.R. is also the recipient of research funding from Merck and Lilly, although those funds were not used in this project. The remaining authors declare no competing interests.
