## [Peer Review File · Nature Communications]

Reviewers' comments:

Reviewer #1 (Remarks to the Author):

This manuscript by Zhang et al reported the interesting and important findings that gut microbial beta-glucuronidases are the key player to cause colitis promotion by Triclosan. The work also clearly presented the detailed molecular mechanism, as evidenced by lots of data covering X-ray crystallography, proteomic analysis, enzyme activity assays, molecular modeling, and LC-MS analysis. This manuscript can be considered for publication after addressing the following issues.

1. Figure 2 demonstrated the conversion of TCS-G to TCS in vitro and in vivo. It is nice that the authors utilized the germ-free mice to examine the conversion. Although the result clearly supported the conclusion, why is there formation of TCS in the colon of germ-free mice (Fig 2H), supposedly there is no gut bacteria existing?

2. Figure 3C studied the GUS activities extracted from human fecal samples. Figure 3D and 3E showed the abundance of total bacterial GUS enzymes and specific types of GUS enzymes, respectively. However, the cited ref (#34) used the rate constant ($1/S$), whereas in this work the authors employed turnover rate ($\mu\text{M}/S$). What is the reason? Is this way to obtain a better correlation (better linear relationship)?

3. However, the authors did not explain the reason why there are limited correlations shown in Fig 3D and 3E. Especially, Fig 3a and 3B already indicated different bacterial GUS enzymes display a wide range of activities.

4. Another regarding Figure 3: The authors initially identified Loop 1 and FMN-binding GUSs are effective to convert TCS-G to TCS. However, according to Fig 3E, only Loop 1 GUS showed the correlation, but not FMN-binding GUS. What is the reason?

5. According previous studies, Loop 1 GUS enzymes cover a number of bacterial enzymes. This is consistent with Fig 3A where the activities of Fp2-L1 GUS and E. coli GUS have a big difference. Therefore the authors need to provide a sequence alignment for at least Loop1 GUS enzymes and then give feasible explanation about the relationship between the loop structure and sequence. Otherwise, it is an oversimplified comparison (FigS6) and the corresponding conclusion (to discuss Fp2-L1 GUS residues, including Y479, M454, M455 and M362) may not be convincing.

Reviewer #2 (Remarks to the Author):

This is a review of a manuscript entitled “Specific Gut Microbial Enzymes Drive Colitis Promotion by Triclosan” authored by Zhang et al. In this manuscript, the authors characterized the molecular mechanisms of TCS toxicity and the role of gut microbiota in the process. Through comprehensive in vitro, ex vivo and in vivo experiments, the authors determined that commensal microbes are responsible for the activation of TCS in the colon and induce gut toxicity and exacerbate inflammation in colitis models. B-glucuronidase (GUS) enzymes and special motifs were identified to generate the active TCS aglycone and targeted inhibition of GUS enzymes reduced the colitis effects of TCS.

The manuscript is of great significance and novelty, it represents a great contribution to the field and sheds some light on the “dark” area of host-microbiome metabolic interactions. The manuscript is well written, and the data is presented clearly.

Major comments:

1) The authors thoroughly describe the LC-MS/MS methods used for their analyses including the ¹³C₁₂-TCS internal standard needed for absolute quantitation in the different experiments. However, it is not clear which isotopically labeled standard they used for TCS-G and TCS-Sulfate if any. This reviewer recognizes the limited availability of isotopically labeled standards and if no isotopically labeled standards were used for the analysis, a separate graph for TCS-G and TCS-Sulfate with AUC or intensity on the y-axis instead of concentration (e.g. pmol/mg tissue) would suffice (Figures 1 C, 2 C, F and H, S1 B and S3 B).

2) In page 12 line 294, the authors state that GUSi has been shown to have no effects on human or mouse intestinal cells or on the activity of mammalian GUS enzymes. Do the authors refer to GUS inhibitors they have used before such as in reference 29 or to the inhibitor used in this study (UNC10201652)? Has the used inhibitor been characterized? If this was described, it was not clear in my reading of the manuscript.

Minor comments:

- 1) Figure 1 F and G are not that informative and could be moved to SI. Figure 1 H is clear enough to show that TCS and TCS-G are the main forms found in stool and urine, respectively.
- 2) Please describe in the methods section how the AUC in Figure 2 E and F was calculated.
- 3) In page 9 line 214 and Figure 3 B, the authors wrote that the TCS-G to TCS conversion was determined by HPLC. In the methods section (Catalytic efficiency assay, page 3), the authors describe the HPLC system but not the detector. It is not clear if MS or other type of detector (UV) was used. Please indicate what detector was used and conditions (e.g. wavelength for UV detector).
- 4) In page 14 line 343, the authors state to have elucidated the molecular mechanism of TSC toxicity. Based on the data presented a more accurate statement would be the activation mechanism of toxicity in the gut (or similar).
- 5) In page 14 line 356, the authors state the number of unique gut microbial GUS enzymes in mice and humans. However, it is important to note that those numbers reflect the known GUS enzymes as many organisms have not been sequenced and the function of every protein has not been fully characterized and annotated. This reviewer suggests not leaving the dark proteome out, as there is still plenty to be discovered.

Reviewer #3 (Remarks to the Author):

Summary:

This study explores the role of triclosan (TCS) oral administration to mice, as and adjuvant factor that worsens DSS induced colitis. Authors have previously published that this chemical has adverse effects on colonic inflammation and associated colon tumorigenesis (PMID: 29848663). In this submission the authors attempt to define a mechanism by which microbes increase toxicity of TCS and suggest inhibition of the enzymes involved in this process enzymatic activity can be a therapeutic approach for IBD.

Major comments:

Relevance of the mouse model and findings to humans is unclear:

a-The authors report in Figure 1E that humans exposed to TCS have ~500 pmol/g of stool total TCS (of which the majority seems to be TCS as seen in figure 1F). In the mouse in vivo experiments the authors report over 10 pmol/mg of TCS in colonic "digesta" (how is digesta defined? Is this colonic

tissue? Colon content? Both?). Thus, the values in humans and mice correspond to different tissues, locations, units. A rough conversion of those units would seem to indicate mice were gavaged with 50 times higher magnitude of TCS than what humans would be exposed to in toothpaste (humans do not receive intragastric bolus of TCS). This raises serious questions about the physiological relevance of the experiments in the paper.

b- At the beginning of the manuscript a point is made regarding TCS presence in several products that humans consume. However, most human subjects recruited had low levels of TCS. This seems to show humans don't have such high levels of TCS, thus suggesting that we might not be consuming that much. Indeed, Figure 1 E shows only 2 human subjects had detectable levels of TCS.

Clinical relevance to IBD is unclear. No evidence to support role of triclosan in increasing IBD incidence and prevalence in the human population.

a-The authors claim that triclosan (TCS) "is an environmental risk factor for IBD and associated diseases" (lines 96 – 99). This statement is not supported by references or strong evidence. I am also not aware of any studies showing that triclosan increases the incidence and prevalence of IBD at a population level.

b- The authors do not show that TCS increases severity of IBD in the human population and therefore, such bold statements should be avoided. For example, lines 342 – 34, the authors claim that "TCS is a potential risk factor of IBD and associated colorectal cancer." This was only shown in mice, not in the human population.

c-The authors claim that this research provides "a new therapeutic approach to alleviate colitis and associated diseases" (Lines 62 – 63). I would argue that removal of TCS would be a better approach. IBD is a multifactorial disease in which genes, a multitude of environmental factors and immune alterations need to combine. It is not caused by TCS, or any single factor, as suggested in lines 351 – 353. These sections require significant editing.

Colitis model. In their previous study the authors used a genetic (IL-10 KO), spontaneous colitis model. Here they use acute 2 % DSS and evaluate histological scores and cytokines after 21 days of high doses of TCS. DSS is a toxic model, that targets the gut epithelial barrier, and could significantly increase the uptake of toxins. A second model of colitis is needed for the experiments testing the inhibitor.

Antibiotic-treated mice. Extensive experimentation is performed using this model. This is problematic as ATB will significantly affect the gut microbiome structure and metabolic activity. The results from these experiments are difficult to interpret, especially given the lack of microbiome analysis in all experiments. Much of the hypothesis that bacteria participate in TCS conversion, rely on these results. ATB treated mice are not "completely depleted" of bacteria (line 170), moreover, bacterial overgrowth of resistant groups will occur. The biochemical pathways leading to first and

second conversion remain unclear. Could dysbiosis induced by ATB prevent the second conversion and indeed what the authors find is accumulation of the first compound in the colon?

Culture exp do not necessarily indicate microbial conversion happens in vivo.

a-It is hypothesized that gut microbiota participates in the conversion of TCS-G to TCS, leading to the accumulation of TCS in the lower gastrointestinal tract. The authors cultured fecal bacteria under anaerobic conditions with TCS to measure TCS-G conversion in vivo. This is a reductionist experiment that could be affected by many factors.

b- I was surprised to see that one of the major microbial metabolizers was *F. prau*. This is a taxon that has consistently been found to be anti-inflammatory and beneficial, and severely depleted in IBD.

c-Can a unique GUS inhibitor inhibit all the GUS enzymes found in human feces? This may well introduce an important bias.

Enzyme inhibitor (GUSi) to prevent the adverse effects of TCS. Wouldn't banning of TCS in toothpaste be a logical step? At least this should be discussed.

Lack of important controls and outcome measurements. Although the enzyme inhibitor (GUSi) seems to be beneficial for DSS-treated mice exposed to TCS, the drug could have several implications in gut physiology. The use of non-DSS treated mice exposed to GUSi is needed to define the adverse effects of the drug in vivo (proinflammatory gene expression, low grade inflammation, etc). GUSi inhibit different microbial GUS, which include beneficial bacteria such as *Faecalibacterium prausnitzii*. What else does GUSi impact in the microbiome?

Other comments.

a) The introduction needs a paragraph explaining TCS metabolism. Where does glucuronidation or/and sulfonation of TCS take place? This is important based on the flow of the story. If mice/humans primarily consume TCG (active), is it possible that some concentrations of the native drug could be influenced by the consumed amount (and non-microbially modified) or intestinal motility? ATB (faster) and germ-free mice (slower) will have significantly different intestinal motility as compared to wild type mice. It is also important to consider that dietary components stay longer in the large intestine than upper in the GI tract.

b- The authors mention reference 22 as a justification to use 16S rRNA gene quantification as a surrogate for bacterial levels. However, the referenced paper focuses on the use of DNA extracted normalized to grams of stool, not 16S rRNA gene quantification.

c- Please define acronyms at first use (FMN, etc).

d- Some sections are a bit repetitive. For instance, line 157-159

Reviewer #1:

This manuscript by Zhang et al reported the interesting and important findings that gut microbial beta-glucuronidases are the key player to cause colitis promotion by Triclosan. The work also clearly presented the detailed molecular mechanism, as evidenced by lots of data covering X-ray crystallography, proteomic analysis, enzyme activity assays, molecular modeling, and LC-MS analysis. This manuscript can be considered for publication after addressing the following issues.

We would like to thank the reviewer for the positive comments. We have addressed all the issues raised by the reviewer, through revising the manuscript (to address point 1, 2, 3, 4, and 5) and performing new data analysis (to address point 3, 4, and 5). The details are discussed below.

1. Figure 2 demonstrated the conversion of TCS-G to TCS in vitro and in vivo. It is nice that the authors utilized the germ-free mice to examine the conversion. Although the result clearly supported the conclusion, why is there formation of TCS in the colon of germ-free mice (Fig 2H), supposedly there is no gut bacteria existing?

We would like to thank the reviewer for this question. The free TCS in the colon of germ-free mice is most likely derived from the small intestine digesta. We showed that after the mice are orally exposed to 80 ppm TCS in diet, the small intestine (SI) digesta contains free TCS, though TCS-G is a major compound (please see Fig. 1). As a result, a mixture of TCS and TCS-G will enter the colon with the flow of digesta. This would lead to appearance of free TCS in the colon tissue of germ-free mice, as we showed in Fig. 2. In the conventional mice, the amount of free TCS in the colon will be further increased due to the gut microbe-mediated conversion of TCS-G to TCS.

Based on reviewer's comment, we have added text in the manuscript to make it clearer:

"We observed the presence of free TCS in the colon of germ-free mice (Fig. 2H), and this is likely derived from the small intestine digesta: we showed that after mice were exposed to 80 ppm TCS in diet, small intestine digesta contains free TCS (Fig. 1A), which could enter the colon with the flow of digesta".

2. Figure 3C studied the GUS activities extracted from human fecal samples. Figure 3D and 3E showed the abundance of total bacterial GUS enzymes and specific types of GUS enzymes, respectively. However, the cited ref (#34) used the rate constant (1/S), whereas in this work the authors employed turnover rate (microM/S). What is the reason? Is this way to obtain a better correlation (better linear relationship)?

We thank the reviewer for this comment. In ref 34, the rate constant was used instead of turnover rate because the authors were able to obtain full turnover progress curves. This was because their substrates SN-38-G and SN-38 have different absorbance profiles and therefore were amenable to detection via a plate reader. Triclosan (TCS) and triclosan-glucuronide (TCS-G) have little difference in their absorbance profiles, forcing us to first quench the reaction using trichloroacetic acid (at five different time points), then separate the analytes using HPLC before detection and quantitation via absorbance. The inherent low throughput nature of the analysis meant that we were limited in the amount of samples we could run and therefore we could only collect initial turnover rates (uM/s) rather than fitting a full progress curve to obtain turnover rate (1/s).

Though correlations using turnover rate versus rate constant would likely have different R^2 values due to slightly different fit methods, the trends would remain the same and therefore the conclusions would not change.

Based on reviewer's comment, we have added the following text in the Materials & Methods section:

"In fimo assay

Reaction mixtures contained 5 μ L fecal extract (0.1 mg/mL final), 30 μ L TCS-G (200 μ M final), and 15 μ L assay buffer (25 mM HEPES, 25 mM NaCl, pH 6.5). Reactions were quenched at five time points with 50 μ L 25% trichloroacetic acid. Samples were centrifuged 20 minutes at 13,000 RPM in a tabletop centrifuge, and supernatant was analyzed via the same HPLC method described for the catalytic efficiency assay. Reaction rates were determined by fitting progress curves using linear regression, and are expressed as initial turnover rates (μ M/s). Controls contained fecal extract that had been heat-killed at 95°C for 5 minutes. Reported rates are the average of three biological replicates \pm SEM".

3. However, the authors did not explain the reason why there are limited correlations shown in Fig 3D and 3E. Especially, Fig 3a and 3B already indicated different bacterial GUS enzymes display a wide range of activities.

4. Another regarding Figure 3: The authors initially identified Loop 1 and FMN-binding GUSs are effective to convert TCS-G to TCS. However, according to Fig 3E, only Loop 1 GUS showed the correlation, but not FMN-binding GUS. What is the reason?

Regarding points 3-4:

We thank the reviewer for this excellent comment. GUS enzymes do display a range of activities *in vitro* as evidenced by cited papers¹⁻⁶ and we observed that Loop 1 and FMN-binding GUS enzymes were particularly efficient at processing TCS-G *in vitro*. This suggested that these two classes were likely responsible for the majority of *in vivo* turnover of TCS-G as well. It therefore was not surprising to find that there was no correlation between TCS-G turnover *in fimo* and the abundances of the other GUS loop classes like Mini loop 1, No loop (non-FMN binding), and all No Loop enzymes.

We were surprised, however, that there was no correlation between FMN-binding GUS enzymes and TCS-G turnover. One potential explanation could be that Loop 1 enzymes vary mainly in a loop region of ~15 residues. FMN-binding GUSs, on the other hand, have variations in the larger C-terminal domain (~150 residues). The effects of these variations are difficult to ascertain using sequence alone: for example, the overall sequence identity between the two fastest FMN-binding processors, *R. hominis* 3 and *R. gnavus* 3, is 52.1%, while the overall sequence identity between the fastest and the slowest FMN-binding enzymes, *R. hominis* 3 and *R. hominis* 2, is 50.9%. The same phenomenon is observed when looking at sequence identities of the C-terminal region alone. Because sequence identity alone was not enough to distinguish fast processors from slow processors, and there is no structural data on this domain, the key differences driving substrate specificity among FMN-binding enzymes remain unknown. It is possible that the abundance of efficient or fast FMN-binding GUS enzymes would correlate with *in fimo* processing rates, but to date, we have no computational method of determining which sequences correspond to fast enzymes and which correspond to slow enzymes.

Based on reviewer's comment, we have revised the manuscript and added the following statement to the Discussion to address this topic:

“Using this strategy, we observed that Loop 1 and FMN-binding GUS enzymes were particularly efficient at processing TCS-G *in vitro*. This result suggests that these two classes were likely responsible for the majority of *in vivo* turnover of TCS-G as well. In support of this notion, using the approach of activity-based probe-enabled proteomics, we found that Loop 1 GUS, but not other classes such as Mini loop 1, No loop (non-FMN binding), and No Loop GUS, is correlated with TCS-G turnover *in fimo*. We were surprised, however, that there was no correlation between FMN-binding GUS enzymes and TCS-G turnover *in fimo*. One potential explanation could be that Loop 1 GUS enzymes vary mainly in the contiguous loop 1 sequence motif, which is only 15-20 residues in length. In contrast, FMN-binding GUS enzymes vary mainly in their large C-terminal domains of ~150 residues in length. To date, no structure of an FMN-binding GUS C-terminal domain has been reported, as they have remained mobile and unresolved in the structures determined thus far. Sequence identity does not appear to be sufficient to distinguish the differences between fast and slow-processing FMN-binding enzymes. For example, the sequence identity between the two fastest FMN-binding processors, Rh3 and *R. gnavus* 3 GUS, is 52.1%, while the sequence identity between the fastest and the slowest FMN-binding enzymes, Rh3 and *R. hominis* 2 GUS, is 50.9%. It is possible that the abundance of efficient or fast FMN-binding GUS enzymes would correlate with *in fimo* TCS-G processing rates; but to date, because of the size of these C-terminal domains of FMN-binding GUS enzymes and our lack of structural knowledge about these domains, the specific motif(s) critical for TCS-G processing remain undefined. Overall, these results support that specific microbial GUS enzymes process TCS-G”.

5. According previous studies, Loop 1 GUS enzymes cover a number of bacterial enzymes. This is consistent with Fig 3A where the activities of Fp2-L1 GUS and *E. coli* GUS have a big difference. Therefore the authors need to provide a sequence alignment for at least Loop1 GUS enzymes and then give feasible explanation about the relationship between the loop structure and sequence. Otherwise, it is an oversimplified comparison (FigS6) and the corresponding conclusion (to discuss Fp2-L1 GUS residues, including Y479, M454, M455 and M362) may not be convincing.

We thank the reviewer for this excellent point. Based on this suggestion, we have updated the manuscript to include a multiple sequence alignment for the Loop 1 region of the GUS enzymes examined to better define how loop sequence relates to activity (**Fig. S9**).

We have added the following text to the manuscript:

“Finally, it is likely that the loop structure of each Loop 1 GUS enzyme plays a key role in substrate processing ability. Unfortunately, this loop remains unresolved in several of the structures resolved to date, making it difficult to elucidate the structural role that this loop plays in substrate recognition. A multiple sequence alignment reveals that there is little sequence identity between the Loop 1 GUS enzymes (**Fig. S9**). For example, even for enzymes that have similar catalytic efficiencies, like *E. eligens* and Fp2-L1 GUS, there are few commonalities in their Loop 1 regions that would allow for correlations to be made between loop structure and enzyme function (**Fig. S9**). Nonetheless, it is still apparent that the presence of a loop at the Loop 1 position appears to be favorable for TCS-G binding when compared to other loop classes”.

Reviewer #2:

This is a review of a manuscript entitled “Specific Gut Microbial Enzymes Drive Colitis Promotion by Triclosan” authored by Zhang et al. In this manuscript, the authors characterized the molecular mechanisms of TCS toxicity and the role of gut microbiota in the process. Through comprehensive in vitro, ex vivo and in vivo experiments, the authors determined that commensal microbes are responsible for the activation of TCS in the colon and induce gut toxicity and exacerbate inflammation in colitis models. B-glucuronidase (GUS) enzymes and special motifs were identified to generate the active TCS aglycone and targeted inhibition of GUS enzymes reduced the colitis effects of TCS.

The manuscript is of great significance and novelty, it represents a great contribution to the field and sheds some light on the “dark” area of host-microbiome metabolic interactions. The manuscript is well written, and the data is presented clearly.

We would like to thank the reviewer for the positive comments. We have addressed all the issues raised by the reviewer, through performing new experiment (to address major point 1) and correcting the references and performing new experiments to characterize GUSi (to address major point 2). In addition, we have also substantially revised the manuscript to address the other points raised by the reviewer. The details are discussed below.

Major comments:

1) The authors thoroughly describe the LC-MS/MS methods used for their analyses including the ¹³C₁₂-TCS internal standard needed for absolute quantitation in the different experiments. However, it is not clear which isotopically labeled standard they used for TCS-G and TCS-Sulfate if any. This reviewer recognizes the limited availability of isotopically labeled standards and if no isotopically labeled standards were used for the analysis, a separate graph for TCS-G and TCS-Sulfate with AUC or intensity on the y-axis instead of concentration (e.g. pmol/mg tissue) would suffice (Figures 1 C, 2 C, F and H, S1 B and S3 B).

We would like to thank the reviewer for this question. The internal standard ¹³C₁₂-TCS was used in this study. No isotopically labeled standards of TCS-G and TCS-Sulfate were used because of the unavailability of corresponding isotopically labeled standards.

Based on reviewer’s suggestion, we performed new experiment and calculated the spike recoveries of the three target compounds in the matrixes of colon digesta. The recoveries of TCS, T S-G, and TCS-Sulfate are shown in Table 1 below.

Table 1: recovery rate of TCS and its metabolites in the matrix of colon digesta
(n = 3 duplicates)

Recovery rate (mean ± SEM)	TCS	TCS-G	TCS-Sulfate
Spiked levels (2 pmol/mg)	101.6 ± 8.9%	91.6 ± 5.4%	95.1 ± 1.4%
Spiked levels (10 pmol/mg)	95.6 ± 3.4%	87.1 ± 5.9%	96.9 ± 6.0%

No significant differences were found among the three compounds. Therefore, ¹³C₁₂-TCS was used for the signal correction of TCS, TCS-G and TCS-Sulfate. It is a strategy for absolute quantitation of analytes when some internal standards are unavailable ^{7,8}.

We acknowledge that this correction may lead to possible bias. To minimize it, for the quantification of TCS, TCS-G and TCS-Sulfate by LC-MS/MS in the different experiments, blank samples from the control group without TCS exposure were used as the matrixes for calibration

curve standards. During the instrumental analysis, the matrix calibration curve was performed at the beginning and at the end of every sample batch. All reported concentrations were determined based on a standard curve with 7-10 data points.

Based on reviewer's comment, we have added the above details in the "Detection of TCS and its metabolites by LC-MS/MS" in the Material and Method section of the manuscript:

"The spike recoveries of the three target compounds in the matrixes of mouse colon digesta were determined. The recoveries (% , mean \pm SEM) were 101.6 ± 8.9 and 95.6 ± 3.4 for TCS, 91.6 ± 5.4 and 87.1 ± 5.9 for TCS-G, 95.1 ± 1.4 and 96.9 ± 6.0 for TCS-Sulfate, based on two spiked levels of 2 pmol/mg and 10 pmol/mg, respectively (n = 3 replicates). No significant differences were found among these three compounds. Therefore, $^{13}\text{C}_{12}$ -TCS was used for the signal correction of TCS, TCS-G and TCS-Sulfate, and it is a strategy for the absolute quantitation of analytes when internal standards are unavailable^{7,8}. For the quantification of TCS, TCS-G and TCS-Sulfate by LC-MS/MS in the different experiments, blank samples from the control group without TCS exposure were used as the matrixes for calibration curve standards. During the instrumental analysis, the matrix calibration curve was performed at the beginning and at the end of every sample batch. All reported concentrations were determined based on a standard curve with 7-10 data points".

2) In page 12 line 294, the authors state that GUSi has been shown to have no effects on human or mouse intestinal cells or on the activity of mammalian GUS enzymes. Do the authors refer to GUS inhibitors they have used before such as in reference 29 or to the inhibitor used in this study (UNC10201652)? Has the used inhibitor been characterized? If this was described, it was not clear in my reading of the manuscript.

As suggested by the reviewer, we have changed the references to the correct citations: a previous publication from our group (*ACS Central Science* 4, 868-879, 2018) has described the effects of GUSi (UNC10201652) on *E. coli* growth and found no effect compared to controls (please see Fig. S12 of this reference). This publication also shows that this inhibitor does not inhibit the activity of bovine GUS enzyme (Fig. S14). In addition, our recent work (*PNAS* 117, 7374-7381, 2020) showed that GUSi has no effect on the proliferation of epithelial cells in the ileum, proximal or distal colon of treated mice (see Fig. S5 of this reference)⁹. We have added these two references to the manuscript.

In addition, we have performed new experiments to further characterize GUSi (UNC10201652):

- (1) We treated mice (not stimulated with DSS) with GUSi, then examined the effects of GUSi on colonic and systematic inflammation (as assessed by body weight, colon length, ELISA analysis of cytokines in plasma, qRT-PCR analysis of gene expression in colon, and colon histology).
- (2) We treated mice (not stimulated with DSS) with GUSi, and performed 16S rRNA sequencing to determine the effects of GUSi on gut microbiota.
- (3) We treated mouse intestinal cells (MC38) and human intestinal cells (HCT-116 and Caco-2) with GUSi and analyzed the effects of GUSi on cell growth *in vitro*.

Our data showed that (i) GUSi treatment had little effects on body weight, colonic or systematic inflammation in mice; (ii) GUSi treatment had little impact on the diversity or composition of gut microbiota in mice; and (iii) treatment with GUSi, at concentration up to 10 μM , had little effect on growth of mouse or human intestinal cells *in vitro* (**Fig. S12-14**). We have added the new data in the manuscript.

We have added the new data in the manuscript and added a paragraph in the Results section:

“After demonstrating that GUSi inhibits GUS-mediated TCS-G processing, we further characterized GUSi. Our previous study showed that GUSi has no effect on growth of *E. coli* or on the activity of mammalian GUS enzyme; deficiency of human GUS results in Sly Syndrome, a potentially fatal lysosomal storage disease¹⁰. In addition, we showed that GUSi has no effect on proliferation of epithelial cells in the ileum, proximal or distal colon of the treated mice⁹. Here we further studied its effects on gut physiology. First, we treated C57BL/6 mice with 1 mg/kg GUSi via oral gavage (a treatment scheme determined from our previous studies^{9,11}) and found that a 3- to 4-week treatment with GUSi had little effects on body weight, colon length, colonic or systematic inflammation, or colon histology in mice (**Fig. S12**). GUSi treatment also had little effect on the diversity or composition of fecal microbiota in mice (**Fig. S13**). Next, we found that a 24-h treatment with GUSi, at a concentration up to 10 μ M, had little effect on growth of mouse or human intestinal cells *in vitro* (**Fig. S14**). Taken together, these results demonstrate that GUSi effectively inhibited GUS-mediated TCS-G processing, with little effect on commensal microbes, mammalian intestinal cells, or mammalian GUS enzyme, supporting that GUSi is highly selective toward the gut microbial GUS enzymes and therefore it is feasible to use GUSi to study the functional roles of microbial GUS enzymes in the gut toxicity of TCS”

Minor comments:

1) Figure 1 F and G are not that informative and could be moved to SI. Figure 1 H is clear enough to show that TCS and TCS-G are the main forms found in stool and urine, respectively.

As suggested by reviewer, we removed Fig. 1F-G to the supplemental section (please see **Fig. S2**).

2) Please describe in the methods section how the AUC in Figure 2 E and F was calculated.

The AUC of each sample was calculated using GraphPad Prism software, Version 9.1.2 (225) (<https://www.graphpad.com/scientific-software/prism/>) with the parameters as following: the baseline is set as Y=0 and the peaks that are less than 10% of distance from minimum to maximum Y are ignored.

We have revised the manuscript and added the details in the Material and Method section of the revised manuscript.

3) In page 9 line 214 and Figure 3 B, the authors wrote that the TCS-G to TCS conversion was determined by HPLC. In the methods section (Catalytic efficiency assay, page 3), the authors describe the HPLC system but not the detector. It is not clear if MS or other type of detector (UV) was used. Please indicate what detector was used and conditions (e.g. wavelength for UV detector).

As suggested, we have updated the Catalytic Efficiency section of the M&M methods to include the following:

“Assay mixtures contained 10 μ L GUS (various final concentrations, between 10-50 nM), 30 μ L TCS-G (final concentrations between 30-120 μ M), and 10 μ L assay buffer (50 mM HEPES, 50 mM NaCl, various pH) or (50 mM sodium acetate, 50 mM NaCl, various pH). Control reactions replaced GUS with buffer. Reactions were quenched at five time points with 50 μ L 25% trichloroacetic acid. Samples were centrifuged for 10 minutes at 13,000 RPM, and the supernatant was subjected to analysis by HPLC on an Agilent 1260 Infinity II system using an Agilent InfinityLab Poroshell 120 C18 column (4.6 x 100 mm, 0.7 μ M particle size). Column temperature was set to 38°C with a flow rate of 0.9 ml/min and injection volume of 40 μ L. Conditions were

set to flow 98% A (water with 0.1% formic acid) and 2% B (acetonitrile with 0.1% formic acid) for two minutes. A linear gradient was then set to flow to 98% B over 10 minutes and held for 4 minutes. Conditions were then ramped down to 98% A for 1 minute and re-equilibrated at 98% A for two minutes. Analytes were detected using an Agilent DAD detector at a wavelength of 280 nm. Concentrations of TCS-G were determined using a standard curve of TCS-G (0-250 μ M). Reaction curves were fit using linear regression, and the resultant initial velocities were plotted against substrate concentration to determine k_{cat}/K_M . Reported catalytic efficiencies are the average of three biological replicates \pm SEM”

4) In page 14 line 343, the authors state to have elucidated the molecular mechanism of TSC toxicity. Based on the data presented a more accurate statement would be the activation mechanism of toxicity in the gut (or similar).

As suggested by reviewer, we have revised this sentence to make it more accurate: “Our recent study showed that exposure to TCS exacerbates colitis in mouse models through gut microbiota-dependent mechanisms⁸. Here we elucidate the molecular mechanisms by which gut microbiota contributes to the metabolic activation and subsequent gut toxicity of TCS”.

In addition, we revised the title of our manuscript to “Microbial Enzymes Induce Colitis by Reactivating Triclosan in the GI Tract” to make this point clearer.

5) In page 14 line 356, the authors state the number of unique gut microbial GUS enzymes in mice and humans. However, it is important to note that those numbers reflect the known GUS enzymes as many organisms have not been sequenced and the function of every protein has not been fully characterized and annotated. This reviewer suggests not leaving the dark proteome out, as there is still plenty to be discovered.

As suggested by reviewer, we have revised the manuscript. In the revised manuscript, we have changed this sentence to “the sequencing data from the Human Microbiome Project suggests that the human and mouse gut microbiotas contain hundreds of unique gut microbial GUS enzymes, which have different substrate specificities varying from small compounds to macromolecules^{12,13}. Novel gut microbial GUS enzymes could be identified from further microbiota sequencing and/or functional characterization”.

Reviewer #3:

Summary:

This study explores the role of triclosan (TCS) oral administration to mice, as and adjuvant factor that worsens DSS induced colitis. Authors have previously published that this chemical has adverse effects on colonic inflammation and associated colon tumorigenesis (PMID: 29848663). In this submission the authors attempt to define a mechanism by which microbes increase toxicity of TCS and suggest inhibition of the enzymes involved in this process enzymatic activity can be a therapeutic approach for IBD.

We would like to thank the reviewer for the comments and suggestions. To address the reviewer's concerns, we have performed new experiments:

- (1) We treated mice with varied doses of TCS (1, 10, and 80 ppm TCS in diet), then determined the concentrations of TCS and its metabolites in mouse gut tissues, then compared with the concentrations of TCS in human stool samples.
- (2) We treated mice (not stimulated with DSS) with GUSi, then examined the effects of GUSi on colonic and systematic inflammation (as assessed by body weight, colon length, ELISA analysis of cytokines in plasma, qRT-PCR analysis of gene expression in colon, and colon histology).
- (3) We treated mice (not stimulated with DSS) with GUSi, and performed 16S rRNA sequencing to determine the effects of GUSi on gut microbiota.
- (4) We treated mouse intestinal cells (MC38) and human intestinal cells (HCT-116 and Caco-2) with GUSi and analyzed the effects of GUSi on cell growth *in vitro*.

The results from the new experiments help to address most issues from the reviewer. The new experiments showed that: (1) at all tested doses (1, 10, and 80 ppm TCS in diet), the metabolic profiles of TCS in mouse gut tissues are the same and are characterized by high abundance of free TCS; (2) at lower doses (1-10 ppm TCS in diet), the concentrations of TCS in mouse gut tissues are comparable to the concentrations of TCS in the gut of TCS-exposed human subjects; (3) GUSi has little effect on colonic or systematic inflammation, or the diversity and composition of gut microbiota in mice; and (4) GUSi has little effects on growth of intestinal cells *in vitro*.

The only remaining issue is the IL-10 KO mouse experiment. The IL-10 KO mouse experiment proposed by the reviewer will help to better understand the roles of gut microbial GUS in the biological effects of TCS. However, there are two concerns to perform this experiment:

- (1) to determine the extent to which co-administration of GUSi attenuates the colitis-enhancing effects of TCS in IL-10 KO mice, we will need to perform long-term (~12-15 weeks) repeated oral gavage to administer GUSi in mice (please see detailed justification of the oral gavage method below). This operation (12-15 weeks of repeated gavage) is prone to cause injury, stress, pain, or death, in mice¹⁴⁻²⁵. Indeed, a recent study showed that repeated oral gavage over 6 weeks caused a mortality rate of ~15% in mice, even performed by an experienced technician¹⁴. Other studies have also shown that repeated oral gavage can cause mortality rates of >50%¹⁵⁻²⁴. In addition, the stress and pain in the mice could confound animal experiment results.
- (2) in our previous study, we showed that TCS exposure exacerbated piroxicam (a nonsteroidal anti-inflammatory drug)-induced colitis model in IL-10 KO mice that were maintained in a specific-pathogen-free (SPF) animal facility (please see Fig. 3 of Ref⁸). This model mimics NSAIDs-induced gut disorders and has some limitations to study the pathogenesis of IBD in humans²⁶. As such, this model also includes a toxin (piroxicam) that physically damages

the gut epithelium, like the DSS model employed in the current manuscript. The IL-10 KO mice develop spontaneous colitis when maintained in a conventional animal facility, and this spontaneous model could better mimic human IBD²⁷. However, it remains unknown whether TCS can exacerbate colitis in the spontaneous IL-10 KO model.

Taken together, because of the difficulties of long-term administration of GUSi in mice and the unknown effect of TCS in spontaneous IL-10 KO mouse model, we did not use the IL-10 KO mouse model in this project.

In this manuscript, we used the DSS-induced colitis model, and we think that the data from this model is sufficient to demonstrate that the gut microbial GUS enzymes are required for the colitis-promoting effects of TCS, because (i) the DSS model is one of the most widely animal models to study colitis²⁸. As the reviewer pointed out, the DSS model induces intestinal barrier dysfunction, which is a common feature of human IBD²⁹. Other IBD models such as the IL-10 KO mouse model also develop gut leakage, though less severe compared with the DSS model³⁰; (ii) our previous research showed that exposure to TCS increased DSS-induced colitis and azoxymethane (AOM)/DSS-induced colitis-associated colon cancer in mice⁸, and (iii) the DSS model is rapid²⁸, allowing us to treat mice with GUSi by oral gavage every other day without causing observable adverse outcomes in the treated mice.

We have added the new experimental data in the manuscript, and substantially revised the manuscript to address the reviewer's points. For the IL-10 KO mouse experiment, we have added text in the discussion to emphasize the importance to perform this experiment in the future. We would hope that these revisions help to address the issues from reviewer.

Major comments:

Relevance of the mouse model and findings to humans is unclear:

a-The authors report in Figure 1E that humans exposed to TCS have ~500 pmol/g of stool total TCS (of which the majority seems to be TCS as seen in figure 1F). In the mouse in vivo experiments the authors report over 10 pmol/mg of TCS in colonic "digesta" (how is digesta defined? Is this colonic tissue? Colon content? Both?). Thus, the values in humans and mice correspond to different tissues, locations, units. A rough conversion of those units would seem to indicate mice were gavaged with 50 times higher magnitude of TCS than what humans would be exposed to in toothpaste (humans do not receive intragastric bolus of TCS). This raises serious questions about the physiological relevance of the experiments in the paper.

We would like to thank the reviewer for this important question. Regarding the dose of TCS used in animal experiment, we have provided detailed justifications in our previous reports^{8,31}. In our previous study, we found that exposure to 10-80 ppm TCS in diet increased the severity of colitis and exacerbated the development of colon tumorigenesis in mouse models, suggesting that TCS could have potential adverse effects on gut health⁸. A critical question is whether the observed effects in the animal experiments could model human exposure to TCS. We have determined the dose (10-80 ppm in diet) based on the following considerations:

- (1) A previous human study has shown that after weeks of daily use of TCS-containing toothpaste, the plasma concentrations of TCS (a combination of TCS plus TCS-G) increased from a baseline of 0.03–2.7 nM to 90–1,000 nM^{32,33}. In our experiment, we found that after the mice were exposed to 10-80 ppm TCS in diet for weeks, the plasma concentration of total TCS (free TCS plus TCS-G) were comparable to those reported in the plasma of TCS-

exposed human subjects⁸. We acknowledge that there could be differences in the metabolism of TCS in humans and mice³⁴, but our approach is a direct comparison of the same analytes in the plasma of human vs. mice.

- (2) The average intake levels of TCS from using consumer products were estimated to be 0.047-0.073 mg/kg/day in humans³⁴ (~0.56-0.88 mg/kg in mice, the dose conversion from humans to mice is calculated as described in Ref³⁵). This dose range is comparable to our doses, notably the lower dose (10 ppm TCS in diet, administering TCS at a dose of ~ 1 mg/kg/day, based on a diet of 3 g daily chow), used in our animal experiments.
- (3) The No-Observed-Adverse-Effect Level (NOAEL) of TCS was reported to be 25-40 mg/kg/day³⁶, which leads to a calculated Acceptable Daily Intake (ADI) of TCS = 0.25-0.4 mg/kg/day³⁷. The ADI dose is comparable to the lower dose (10 ppm in diet, ~ 1 mg/kg/day) used in our animal experiment.
- (4) We performed a short-time (several weeks) treatment, it is possible that long-term exposure to TCS at lower doses might also induce adverse effects on gut health.

Based on these considerations, in our previous study we used a dose regime of 10-80 ppm in diet to study the potential gut toxicity of TCS^{8,31}. We acknowledge that there are many challenges to use animal models to study human exposure to consumer antimicrobials such as TCS, since the accurate assessment of exposure and absorption of TCS in human populations is largely unknown and there could be substantial inter-individual variations in exposure and metabolism to TCS³⁸.

In our first submission of this manuscript, we treated mice with 80 ppm TCS in diet. To address reviewer's comments, we performed new experiments: we treated mice with varied doses of TCS (1, 10, and 80 ppm TCS in diet) for 4 weeks, then used LC-MS/MS to measure the concentrations of TCS and its metabolites in gut. The results from the new experiments showed that:

- (1) At all tested doses (1, 10, and 80 ppm TCS in diet), the metabolic profiles of TCS in the gut tissues are the same and are characterized by high abundance of free TCS: ~94-100% of detected TCS species in gut sections, including colon digesta, cecum digesta, and feces, were present as free TCS.
- (2) After the mice were exposed to TCS, notably at the lower doses (1 and 10 ppm in diet), the concentrations of TCS in mouse gut tissues are comparable or within several folds of the concentrations observed in the stool samples of TCS-exposed human subjects: the concentrations of free TCS in mouse colon digesta was 1.5 and 14.7 pmol/mg tissue after exposed to 1 and 10 ppm TCS in diet, respectively (**Fig. 1**); in comparison, the concentration of free TCS in human stool can reach up to ~1 pmol/mg tissue (**Table S1**). This result supports the notion that it is feasible to use animal experiments to model human exposure to TCS, though we acknowledge that there are many challenges to use animal models to study human exposure to consumer chemicals such as TCS.

In this manuscript, the "colon digesta" is the content isolated from the colon tissue. We analyzed this compartment since it has direct interactions with the gut microbes. "Colon mucosa" shown in the manuscript is the colon tissue.

Based on reviewer's comment, we have added the new data in the manuscript (**Fig. 1B**), and added a paragraph in the Discussion section:

"Previous research regarding the metabolism of TCS, as well as many other environmental compounds, has focused on the metabolic processes in mammalian host tissues (e.g., liver), while their metabolic fates in the gut tissues are not well characterized^{39,40}. Here we showed that after TCS exposure in mice, the dominant compound in most host tissues is its conjugated metabolites such

as TCS-G, akin to that reported previously^{39,40}; however, the dominant compound in gut is free TCS. We treated mice with varied doses of TCS (1, 10, and 80 ppm TCS in diet) and found that at all tested doses, the gut tissues had similar metabolic profiles of TCS and were dominated by free TCS. Additionally, we found that after the mice were exposed to TCS, notably at the lower doses (1 and 10 ppm in diet), the concentrations of TCS in mouse gut tissues are comparable or within several folds of the concentrations of TCS observed in the stool of TCS-exposed human subjects (see mouse data in **Fig. 1B** and human data in **Table S1**). This result supports that it is feasible to use animal experiments to model human exposure to TCS, though we acknowledge that there are many challenges to use mouse models to study human exposure to consumer chemicals such as TCS. In addition, we found that after TCS exposure in humans, the human stool samples also exhibited the same TCS metabolic profile as we observed in the animal experiments and contained a high abundance of free TCS. Taken together, these results support that compared with other organs, the gut tissue has a unique profile of TCS metabolism”.

b- At the beginning of the manuscript a point is made regarding TCS presence in several products that humans consume. However, most human subjects recruited had low levels of TCS. This seems to show humans don't have such high levels of TCS, thus suggesting that we might not be consuming that much. Indeed, Figure 1 E shows only 2 human subjects had detectable levels of TCS.

We utilized urine and stool samples from a previous study published by Dr. Parsonnet who is a co-author of this manuscript⁴¹: in this study, healthy volunteers were recruited, subjected to a washout period of at least 16 days (no usage of TCS-containing products), then randomly assigned to two groups which used personal care products with or without TCS for up to 4 months (please see scheme of experiment in **Fig. 1C** and detailed description of the human study in Ref⁴¹).

Based on the experimental design, at t = 0 month (after the washout period), we would expect that most human subjects have little TCS detected in tissues. And this is the case, except two human subjects which showed detectable TCS, likely due to ubiquitous nature of TCS in the environment (please see t= 0 month in **Fig. S2A**). After human subjects were exposed to TCS-containing products for 1-4 months, the concentrations of TCS increased to up to 1 pmol/mg (~1,000 nM) in stool samples (please see t = 1-4 month in **Fig. S2A**, and the complete raw data in Table S1-2).

Overall, these data suggest that (i) the low concentrations of TCS at t = 0 month are expected due to the washout period; and (ii) after even 1 month of routine exposure to TCS through using consumer products, TCS can reach the gut tissues and the concentrations in the gut can be high.

As suggested by reviewer, we have revised the manuscript: we revised Fig. 1 to show a clearer scheme of human experiment and revised the manuscript in the Results section to make it clearer to the readers.

Clinical relevance to IBD is unclear. No evidence to support role of triclosan in increasing IBD incidence and prevalence in the human population.

a-The authors claim that triclosan (TCS) “is an environmental risk factor for IBD and associated diseases” (lines 96 – 99). This statement is not supported by references or strong evidence. I am also not aware of any studies showing that triclosan increases the incidence and prevalence of IBD at a population level.

b- The authors do not show that TCS increases severity of IBD in the human population and therefore, such bold statements should be avoided. For example, lines 342 – 34, the authors claim that “TCS is a potential risk factor of IBD and associated colorectal cancer.” This was only shown in mice, not in the human population.

We would like to thank the reviewer for this comment. We agree with the reviewer on this point. To date, there are only animal experiments which showed that exposure to TCS could potentially increase colitis and colon tumorigenesis in mouse models ⁸.

As suggested by reviewer, we have revised the manuscript to make it more accurate. For example, in the abstract, we have changed the first sentence from the old version of “Triclosan (TCS), an antimicrobial agent in thousands of consumer products, is a risk factor for colitis and colitis-associated colorectal cancer” to “**Emerging research supports that triclosan (TCS), an antimicrobial agent in thousands of consumer products, exacerbates colitis and colitis-associated colorectal tumorigenesis in animal models**”. In the revised introduction section, we have also clearly stated that “**This finding supports that TCS could be a potential risk factor for IBD and associated diseases though further studies are needed to determine its impacts in human populations**”.

c-The authors claim that this research provides “a new therapeutic approach to alleviate colitis and associated diseases” (Lines 62 – 63). I would argue that removal of TCS would be a better approach. IBD is a multifactorial disease in which genes, a multitude of environmental factors and immune alterations need to combine. It is not caused by TCS, or any single factor, as suggested in lines 351 – 353. These sections require significant editing.

We would like to thank the reviewer for this comment. We agree with the reviewer and have revised the manuscript according to the reviewer’s comment. We have deleted all the sentences which stated that GUSi could be used as a potential therapeutic approach.

In addition, we emphasized in this manuscript that we used GUSi as a chemical probe to study the molecular mechanism of TCS. Please see the revised Discussion section, copied below:

“Because genetic tools that specifically target gut microbial GUS enzymes are sparse ^{11,42}, we used a pharmacological approach and employed GUSi as a chemical probe to elucidate the molecular mechanisms of TCS ^{2,43}”.

Colitis model. In their previous study the authors used a genetic (IL-10 KO), spontaneous colitis model. Here they use acute 2 % DSS and evaluate histological scores and cytokines after 21 days of high doses of TCS. DSS is a toxic model, that targets the gut epithelial barrier, and could significantly increase the uptake of toxins. A second model of colitis is needed for the experiments testing the inhibitor.

We would like to thank the reviewer for this comment. The experiment proposed by the reviewer will help us to better understand the roles of gut microbial GUS enzymes in the colitis-enhancing effects of TCS. However, there are some practical reasons why we did not use the IL-10 KO mouse model in this project:

- (1) to determine the extent to which co-administration of GUSi attenuates the colitis-enhancing effects of TCS in IL-10 KO mice, we will need to perform long-term (~12-15 weeks) repeated oral gavage to administer GUSi in mice (please see detailed justification of the oral gavage

method below). This operation (12-15 weeks of repeated gavage) is prone to cause injury, stress, pain, or death, in mice ¹⁴⁻²⁵. Indeed, a recent study showed that repeated oral gavage over 6 weeks caused a mortality rate of ~15% in mice, even performed by an experienced technician ¹⁴. Other studies have also shown that repeated oral gavage can cause mortality rates of >50% ¹⁵⁻²⁴. In addition, the stress and pain in the mice could confound animal experiment results.

- (2) in our previous study, we showed that TCS exposure exacerbated piroxicam (a nonsteroidal anti-inflammatory drug)-induced colitis model in IL-10 KO mice that were maintained in a specific-pathogen-free (SPF) animal facility (please see Fig. 3 of Ref ⁸). This model mimics NSAIDs-induced gut disorders and has some limitations to study the pathogenesis of IBD in humans ²⁶. As such, this model also includes a toxin (piroxicam) that physically damages the gut epithelium, like the DSS model employed in the current manuscript. The IL-10 KO mice develop spontaneous colitis when maintained in a conventional animal facility, and this spontaneous model could better mimic human IBD ²⁷. However, it remains unknown whether TCS can exacerbate colitis in the spontaneous IL-10 KO model.

The details are discussed below:

The IL-10 KO mice take a long time to develop colitis: previous studies showed that at an age of ~20 weeks, the IL-10 KO mice developed colitis ^{44,45}. This is consistent with what we observed: in our previous research ^{8,46,47}, we treated piroxicam-induced specific-pathogen-free (SPF) IL-10 KO mice or IL-10 KO mice maintained in a conventional animal facility (starting age = 6-8 weeks) with TCS or triclocarban (TCC, an antimicrobial compound similar to TCS) via diet for 12-15 weeks, and found that (i) at an age of ~20 weeks, the IL-10 KO mice developed colitis, and (ii) compared with vehicle control, treatment with TCS or TCC increased the severity of colitis in IL-10 KO mice ^{8,46,47}. Therefore, to use the IL-10 KO mouse model to determine the roles of gut microbial GUS enzymes in the colitis-enhancing effects of TCS, we will need to treat the mice with TCS, with or without co-administration of GUSi, for long period (~12-15 weeks, a time interval determined from previous research ^{8,46,47}).

To administer GUSi to inhibit gut microbial GUS enzymes *in vivo*, we will need to use the administration method of oral gavage, because: (i) our previous studies have shown that it requires daily oral gavage of GUSi to effectively inhibit gut microbial GUS enzymes ^{9,11}, (ii) in this manuscript, we treated basal or DSS-induced mice with GUSi by oral gavage (see **Fig. 5**, **Fig. S12**, and **Fig. S13**). We would like to keep the administration method of GUSi consistent in this manuscript, otherwise it would be difficult to interpret the experimental results; and (iii) it is difficult to administer GUSi by other methods, such as dissolving in drinking water since GUSi is poorly solubilized in water; and (iv) there are some recently developed voluntary oral administration methods ⁴⁸, but it will require substantial validation experiments to test whether GUSi can be administered by these methods and whether the administered GUSi inhibits gut microbial GUS enzymes *in vivo*.

Long-term (~12-15 weeks) repeated oral gavage is prone to cause injury, stress, pain, and even death in the mice. Indeed, a recent study showed that repeated oral gavage over 6 consecutive weeks caused a mortality rate of ~15% in CD-1 mice, even performed by an experienced technician ¹⁴. Other studies have also shown that repeated oral gavage can cause many problems from stress to death in the animals: several studies reported mortality rates of >50% ¹⁵⁻²⁴. In addition, the stress and pain in the mice induced by repeated gavages could potentially confound experimental results, especially considering that the IL-10 KO mice develop a mild phenotype of colitis ^{8,44-47}.

In addition, our previous study showed that TCS exposure increased piroxicam-induced colitis in specific-pathogen-free (SPF) IL-10 KO mice⁸. This model (piroxicam-induced colitis in SPF IL-10 KO mice) mimics nonsteroidal anti-inflammatory drugs (NSAIDs)-induced gut disorders and has limitations to study the pathogenesis of colitis in humans²⁶. The IL-10 KO mice develop spontaneous colitis when maintained in a conventional animal facility and the spontaneous IL-10 KO model can better mimic human IBD. However, it remains unknown whether TCS can exacerbate colitis in the spontaneous IL-10 KO model.

Taken together, because of the difficulties of long-term administration of GUSi and the unknown effect of TCS in spontaneous IL-10 KO mouse model, we did not use the IL-10 KO mouse model in this project. We expect that discovery of new classes of GUSi compounds and/or development of new drug formation methods will help us to perform long-term administration of GUSi in animals.

We used the DSS-induced colitis model in this project, because (i) the DSS model is one of the most widely model to study colitis²⁸. As the reviewer pointed out, the DSS model induces intestinal barrier dysfunction, which is a common feature of IBD and its associated diseases²⁹. Other IBD models such as the IL-10 KO mouse model also develop gut leakage, though less severe compared with the DSS model³⁰; (ii) our previous research showed that exposure to TCS increased DSS-induced colitis and azoxymethane (AOM)/DSS-induced colitis-associated colon cancer in mice⁸, and (iii) the DSS model is rapid²⁸, allowing us to treat mice with GUSi by oral gavage every other day without causing observable adverse outcomes in treated mice (see scheme of animal experiment in **Fig. 5**).

Based on reviewer comment, we have revised the manuscript and added text in the discussion:

“Besides the DSS-induced colitis model, our previous study showed that TCS exposure exacerbated piroxicam-induced colitis in specific-pathogen-free (SPF) *Il-10*^{-/-} mice⁸. The conventionally housed *Il-10*^{-/-} mice develop spontaneous colitis, and this spontaneous model can better model human IBD compared with the piroxicam-induced colitis model in SPF *Il-10*^{-/-} mice²⁷. It would be important to determine whether TCS exposure exacerbates colitis in the spontaneous *Il-10*^{-/-} model and to elucidate the extent to which microbial GUS enzymes contribute to the biological effects of TCS in the spontaneous *Il-10*^{-/-} model.”

Antibiotic-treated mice. Extensive experimentation is performed using this model. This is problematic as ATB will significantly affect the gut microbiome structure and metabolic activity. The results from these experiments are difficult to interpret, especially given the lack of microbiome analysis in all experiments. Much of the hypothesis that bacteria participate in TCS conversion, rely on these results. ATB treated mice are not “completely depleted” of bacteria (line 170), moreover, bacterial overgrowth of resistant groups will occur. The biochemical pathways leading to first and second conversion remain unclear. Could dysbiosis induced by ATB prevent the second conversion and indeed what the authors find is accumulation of the first compound in the colon?

Culture exp do not necessarily indicate microbial conversion happens in vivo. a-It is hypothesized that gut microbiota participates in the conversion of TCS-G to TCS, leading to the accumulation of TCS in the lower gastrointestinal tract. The authors cultured fecal bacteria under anaerobic conditions with TCS to measure TCS-G conversion in vivo. This is a reductionist experiment that could be affected by many factors.

We would like to thank the reviewer for this excellent question and we agree with the reviewer that each approach (e.g. antibiotic suppression of gut microbiota or *in vitro* culture of bacteria)

has its limitations. To increase the rigor of our research, in this manuscript we have used multiple complementary techniques, to aid in avoiding weakness associated with each technique. Specially, to determine the roles of gut microbes and gut microbial enzymes involved TCS metabolism, we used a combination of several different approaches, including:

- (1) LC-MS/MS analysis of TCS and its metabolites along the gastrointestinal tract (**Fig. 1**)
- (2) *in vitro* culturing studies using fecal bacteria from mice or humans (**Fig. 2A**)
- (3) antibiotic-mediated suppression of gut bacteria *in vivo*: we performed both long-term treatment (**Fig. 2B-C**) and short-time time-course study (**Fig. 2D-F**)
- (4) germ-free mice established on C57BL/6 background (**Fig. 2G-H**)
- (5) germ-free mice established on Swiss Webster background (**Fig. S5**)
- (6) enzymatic assays using purified gut microbial GUS enzymes (**Fig. 3A-B**)
- (7) enzymatic assays and proteomics using human fecal samples (**Fig. 3C-E**)

We showed that: (i) the concentration of TCS increased, while the concentration of TCS-G decreased, from the proximal to the distal regions of the intestinal tract (**Fig. 1**), suggesting a potential conversion of TCS-G to TCS in the colon; (ii) fecal bacteria from mice or humans, purified gut microbial GUS enzymes, and human fecal samples can catalyze the conversion of TCS-G to TCS *in vitro* (**Fig. 2A, Fig. 3A-B, Fig. 3C-E**). This result suggests that gut microbes or gut microbial enzymes can catalyze the de-glucuronidation reaction to convert TCS-G to TCS; and (iii) suppression of gut microbiota, using antibiotic or germ-free approaches, increased TCS-G while decreased TCS in colon digesta in mice (**Fig. 2B-H and Fig. S5**), suggesting that gut microbes contribute to the colonic conversion of TCS-G to TCS *in vivo*. Taken together, these results support our hypothesis that gut microbes or gut microbial enzymes converts TCS-G to TCS.

For our antibiotic experiment, we have analyzed gut microbial abundance in mice and showed that the antibiotic treatment dramatically reduce the total gut microbial abundance (**Fig. S4**), supporting the microbiota-suppressing effects of the antibiotic, and this finding is consistent with previous findings using the same antibiotic composition^{47,49-51}. In addition, we also performed germ-free mouse experiment, using germ-free mice established on C57BL/6 background (**Fig. 2G-H**) and Swiss Webster background (**Fig. S5**). The results obtained from the antibiotic and germ-free approaches are consistent, supporting our hypothesis that gut microbes converts TCS-G to TCS.

Overall, our finding is consistent with previous research regarding the gut microbial metabolism of glucuronidated conjugates. Previous study has shown that many xenobiotics are metabolized in the liver to generate glucuronidated conjugates, which are excreted in the bile and enter the duodenum, then subjected to bacterial de-glucuronidation in the colon tissues⁴².

As suggested by reviewer, we have revised the manuscript:

- (1) In the result section, we revised the manuscript to emphasize that we used a combination of many complementary approaches, not only the *in vitro* gut bacteria culture experiment, to test our hypothesis: “To test this hypothesis, we used a combination of approaches including *in vitro* culturing of gut bacteria, antibiotic-mediated suppression of gut bacteria *in vivo*, and germ-free mice to examine the roles of the gut microbiota in colonic metabolism of TCS”.
- (2) We changed the sentence of “this cocktail effectively depleted gut bacteria in mice” to “this cocktail effectively reduced gut bacteria in mice”. This statement is a more accurate term to describe the impact of antibiotic treatment on gut microbiota.
- (3) In the discussion section, we have provided extra discussion of our findings “Using a combination of approaches including *in vitro* culturing of gut bacteria, antibiotic-mediated suppression of gut bacteria *in vivo*, and germ-free mice, we found that gut microbiota

convert TCS-G to TCS in the colon and therefore contribute to the unique metabolic profile of TCS in the colon. Overall, these results support a model that after TCS exposure, it is metabolized in host tissues (notably the liver) and is converted to the conjugated metabolites such as TCS-G, which are then released to the intestines and are subjected to bacterial de-glucuronidation in colon⁴². Other gastrointestinal factors, such as intestinal mobility and food intake, have been shown to modulate drug pharmacokinetics^{52,53}, and these factors could also affect the metabolic fates of TCS in gut”.

b- I was surprised to see that one of the major microbial metabolizers was *F. prausnitzii*. This is a taxon that has consistently been found to be anti-inflammatory and beneficial, and severely depleted in IBD.

Thank you for this comment. We did include two GUS isoforms from *F. prausnitzii* in our panel of *in vitro* enzymes because this taxon is very common in the gut microbiome, making up between 5-15% of the gut bacterial population⁵⁴. It is not surprising that both of these enzymes are efficient processors of TCS-G, as they both are Loop 1 enzymes and have 75% sequence identity to each other. Our panel additionally shows that enzymes from other taxa also are highly efficient at processing TCS-G, however, so it would not be appropriate to assign all responsibility for TCS-G turnover to this one taxon. Even though *F. prausnitzii* is depleted in IBD, other GUS-containing taxa may still be present that can process this substrate. Certainly, this observation demonstrates the importance of taking into account interpersonal gut microbiota variations when considering the effects that TCS may have on different people.

c-Can a unique GUS inhibitor inhibit all the GUS enzymes found in human feces? This may well introduce an important bias.

Pan-GUS inhibitors do exist, including compounds like D-glucaro-1,4-lactone⁵⁵, D-glucaro-1,5-lactone⁵⁶, and uronic-neurostegine⁵⁷. However, the inhibitor used in this study (GUSi) has been shown to selectively target Loop 1 GUSs over other loop classes in other literature^{56,58} and now in this work, FMN-binding GUSs as well. Use of a pan-GUS inhibitor in mammals is not possible as inhibition of mammalian GUS enzymes leads to Sly syndrome, a potentially fatal lysosomal storage disease. However, a pan-GUS inhibitor could very well be used as a chemical tool for *in vivo* assays.

Enzyme inhibitor (GUSi) to prevent the adverse effects of TCS. Wouldn't banning of TCS in toothpaste be a logical step? At least this should be discussed.

We would like to thank the reviewer for this comment. We agree with the reviewer and have revised the manuscript according to the reviewer's comment. We have deleted all the sentences which stated that GUSi could be used as a potential therapeutic approach, and emphasized that in this manuscript we used GUSi as a chemical probe to study the molecular mechanism of TCS. In addition, in the end of the manuscript, we emphasized that it is important to re-evaluate the usage of TCS in products: “They also suggest that the safety of TCS and related compounds should be reconsidered given their potential for intestinal damage”.

Lack of important controls and outcome measurements. Although the enzyme inhibitor (GUSi) seems to be beneficial for DSS-treated mice exposed to TCS, the drug could have several implications in gut physiology. The use of non-DSS treated mice exposed to GUSi is needed to define the adverse effects of the drug *in vivo* (proinflammatory gene expression, low grade inflammation, etc). GUSi inhibit different microbial GUS, which include beneficial bacteria such as *Faecalibacterium prausnitzii*. What else does GUSi impact in the microbiome?

We would like to thank the reviewer for this question. A previous publication from our group (*ACS central science* 4, 868-879, 2018) has described the effects of GUSi (UNC10201652) on *E. coli* growth and found no effect compared to controls (see Fig. S12 of this reference). This publication also shows that this inhibitor does not inhibit bovine GUS (Fig. S14). In addition, our

recent work (*Proc Natl Acad Sci U S A* **117**, 7374-7381, 2020) showed that GUSi (UNC10201652) has no effect on the proliferation of epithelial cells in the ileum, proximal or distal colon of treated mice (see Fig. S5 of this reference)⁹. We have cited these two papers in the manuscript.

In addition, to further address reviewer's comments, we have performed new experiments to further characterize the GUSi (UNC10201652):

- (1) We treated mice (not stimulated with DSS) with GUSi, then examined the impact of GUSi on colonic and systematic inflammation (as assessed by ELISA of cytokines in plasma, qRT-PCR analysis of gene expression in colon, and colon histology);
- (2) We treated mice (not stimulated with DSS) with GUSi, and performed 16S rRNA sequencing to determine the effects of GUSi on gut microbiota.
- (3) We treated mouse intestinal cells (MC38) and human intestinal cells (HCT-116 and Caco-2) with GUSi and analyzed the effects of GUSi on intestinal cell growth.

Our data showed that (i) GUSi treatment had little effects on colonic or systematic inflammation in mice, (ii) GUSi treatment had little impact on the diversity or composition of gut microbiota in mice; and (iii) GUSi had little effect on growth of intestinal cells *in vitro*. Together with our previous studies^{9,10}, these results demonstrate that GUSi had little effect on commensal microbes, mammalian intestinal cells, or mammalian GUS enzyme, supporting that it is feasible to use GUSi to study the functional roles of microbial GUS enzymes in the gut toxicity of TCS.

We have added the new data in the manuscript and added a paragraph in the Results section to describe the new data:

“After demonstrating that GUSi inhibits GUS-mediated TCS-G processing, we further characterized GUSi. Our previous study showed that GUSi has no effect on growth of *E. coli* or on the activity of mammalian GUS enzyme; deficiency of human GUS results in Sly Syndrome, a potentially fatal lysosomal storage disease¹⁰. In addition, we showed that GUSi has no effect on proliferation of epithelial cells in the ileum, proximal or distal colon of the treated mice⁹. Here we further studied its effects on gut physiology. First, we treated C57BL/6 mice with 1 mg/kg GUSi via oral gavage (a treatment scheme determined from our previous studies^{9,11}) and found that a 3- to 4-week treatment with GUSi had little effects on body weight, colon length, colonic or systematic inflammation, or colon histology in mice (**Fig. S12**). GUSi treatment also had little effect on the diversity or composition of fecal microbiota in mice (**Fig. S13**). Next, we found that a 24-h treatment with GUSi, at a concentration up to 10 μ M, had little effect on growth of mouse or human intestinal cells *in vitro* (**Fig. S14**). Taken together, these results demonstrate that GUSi effectively inhibited GUS-mediated TCS-G processing, with little effect on commensal microbes, mammalian intestinal cells, or mammalian GUS enzyme, supporting that GUSi is highly selective toward the gut microbial GUS enzymes and therefore it is feasible to use GUSi to study the functional roles of microbial GUS enzymes in the gut toxicity of TCS”

Other comments.

a) The introduction needs a paragraph explaining TCS metabolism. Where does glucuronidation or/and sulfonation of TCS take place? This is important based on the flow of the story.

As suggested by reviewer, we have added text in the Introduction section to explain TCS metabolism:

“Previous studies have shown that once TCS enters the body, it is rapidly metabolized in host tissues, such as liver, to form the glucuronide-conjugated metabolite TCS-glucuronide (TCS-G), which is biologically inactive and is thought to be quickly eliminated from the body^{39,40}. Given this rapid metabolic inactivation, though, it has remained unclear how exposure to low-dose TCS causes gut toxicity *in vivo*. We hypothesize that gut microbial enzymes act on key TCS metabolites in the colon, leading to unique gut metabolic profiles highlighted by reactivation of TCS in the gut and resulting in subsequent gut toxicology”.

If mice/humans primarily consume TCG (active), is it possible that some concentrations of the native drug could be influenced by the consumed amount (and non-microbially modified) or intestinal motility? ATB (faster) and germ-free mice (slower) will have significantly different intestinal motility as compared to wild type mice. It is also important to consider that dietary components stay longer in the large intestine than upper in the GI tract.

Previous research supports that intestinal mobility, as well as food intake, have impacts on pharmacokinetics of orally administered drugs^{52,53}. It is feasible that these factors could also affect the metabolic fates of TCS. A recent study have shown that TCS is found in some commonly consumed food products⁵⁹, suggesting potential food-TCS interactions. To our best knowledge, few studies have investigated the impacts of these gastrointestinal factors on metabolic fates of TCS.

As suggested by reviewer, we have added text in the Discussion section:

“Using a combination of approaches including *in vitro* culturing of gut bacteria, antibiotic-mediated suppression of gut bacteria *in vivo*, and germ-free mice, we found that gut microbiota convert TCS-G to TCS in the colon and therefore contribute to the unique metabolic profile of TCS in the colon. Overall, these results support a model that after TCS exposure, it is metabolized in host tissues (notably the liver) and is converted to the conjugated metabolites such as TCS-G, which are then released to the intestines and are subjected to bacterial de-glucuronidation in colon⁴². Other gastrointestinal factors, such as intestinal mobility and food intake, have been shown to modulate drug pharmacokinetics^{52,53}, and these factors could also affect the metabolic fates of TCS in gut”.

b- The authors mention reference 22 as a justification to use 16S rRNA gene quantification as a surrogate for bacterial levels. However, the referenced paper focuses on the use of DNA extracted normalized to grams of stool, not 16S rRNA gene quantification.

We would like to thank the reviewer for this question and have corrected the reference. In the revised manuscript, we have changed the reference to: “Vijay-Kumar, M., et al. Metabolic syndrome and altered gut microbiota in mice lacking Toll-like receptor 5. *Science* (New York, N.Y.) 328, 228-231 (2010)”. This reference used the same antibiotic composition (1.0g/L ampicillin and 0.5g/L neomycin) as used in our study, and used 16S rRNA gene as a marker to measure the effect of antibiotic treatment on fecal microbiota (Fig. S13).

c- Please define acronyms at first use (FMN, etc).

We have added the full name of FMN “flavin mononucleotide” and HPLC “high-performance liquid chromatography” at the first use in the manuscript. In addition, we performed a detailed proofreading of the manuscript.

d- Some sections are a bit repetitive. For instance, line 157-159

Based on reviewer’s suggestion, we have deleted the repetitive sentence and changed this section to “The data presented above revealed that the concentration of TCS increased, while the

concentration of TCS-G decreased, from the proximal to the distal regions of the intestinal tract (Fig. 1A)".

References

1. Bhatt, A.P., *et al.* Targeted inhibition of gut bacterial β -glucuronidase activity enhances anticancer drug efficacy. *Proceedings of the National Academy of Sciences* **117**, 7374-7381 (2020).
2. Biernat, K.A., *et al.* Structure, function, and inhibition of drug reactivating human gut microbial β -glucuronidases. *Scientific reports* **9**, 1-15 (2019).
3. Pellock, S.J., *et al.* Discovery and characterization of FMN-binding β -glucuronidases in the human gut microbiome. *Journal of molecular biology* **431**, 970-980 (2019).
4. Ervin, S.M., *et al.* Targeting regorafenib-induced toxicity through inhibition of gut microbial β -glucuronidases. *ACS chemical biology* **14**, 2737-2744 (2019).
5. Ervin, S.M., *et al.* Gut microbial β -glucuronidases reactivate estrogens as components of the estrobolome that reactivate estrogens. *Journal of Biological Chemistry* **294**, 18586-18599 (2019).
6. Pellock, S.J., *et al.* Three structurally and functionally distinct β -glucuronidases from the human gut microbe *Bacteroides uniformis*. *Journal of Biological Chemistry* **293**, 18559-18573 (2018).
7. Liao, C. & Kannan, K. Determination of free and conjugated forms of bisphenol A in human urine and serum by liquid chromatography-tandem mass spectrometry. *Environ Sci Technol* **46**, 5003-5009 (2012).
8. Yang, H., *et al.* A common antimicrobial additive increases colonic inflammation and colitis-associated colon tumorigenesis in mice. *Sci Transl Med* **10**(2018).
9. Bhatt, A.P., *et al.* Targeted inhibition of gut bacterial beta-glucuronidase activity enhances anticancer drug efficacy. *Proc Natl Acad Sci U S A* **117**, 7374-7381 (2020).
10. Pellock, S.J., *et al.* Gut Microbial beta-Glucuronidase Inhibition via Catalytic Cycle Interception. *ACS central science* **4**, 868-879 (2018).
11. Wallace, B.D., *et al.* Alleviating cancer drug toxicity by inhibiting a bacterial enzyme. *Science (New York, N.Y.)* **330**, 831-835 (2010).
12. Pollet, R.M., *et al.* An atlas of β -glucuronidases in the human intestinal microbiome. *Structure* **25**, 967-977. e965 (2017).
13. Creekmore, B.C., *et al.* Mouse Gut Microbiome-Encoded β -Glucuronidases Identified Using Metagenome Analysis Guided by Protein Structure. *MSystems* **4**, e00452-00419 (2019).
14. Arantes-Rodrigues, R., *et al.* The effects of repeated oral gavage on the health of male CD-1 mice. *Lab Animal* **41**, 129-134 (2012).
15. Murphy, S.J., Smith, P., Shaivitz, A.B., Rossberg, M.I. & Hurn, P.D. The effect of brief halothane anesthesia during daily gavage on complications and body weight in rats. *Contemp Top Lab Anim Sci* **40**, 9-12 (2001).
16. Brown, A.P., Dinger, N. & Levine, B.S. Stress produced by gavage administration in the rat. *Contemp Top Lab Anim Sci* **39**, 17-21 (2000).
17. Balcombe, J.P., Barnard, N.D. & Sandusky, C. Laboratory routines cause animal stress. *Contemp Top Lab Anim Sci* **43**, 42-51 (2004).
18. Murphy, S.J., Smith, P., Shaivitz, A.B., Rossberg, M.I. & Hurn, P.D. The effect of brief halothane anesthesia during daily gavage on complications and body weight in rats. *Journal of the American Association for Laboratory Animal Science* **40**, 9-12 (2001).
19. Brown, A.P., Dinger, N. & Levine, B.S. Stress produced by gavage administration in the rat. *Journal of the American Association for Laboratory Animal Science* **39**, 17-21 (2000).

20. Balcombe, J.P., Barnard, N.D. & Sandusky, C. Laboratory routines cause animal stress. *Journal of the American Association for Laboratory Animal Science* **43**, 42-51 (2004).
21. Germann, P.-G., Ockert, D. & Tuch, K. Oropharyngeal granulomas and tracheal cartilage degeneration in Fischer-344 rats. *Toxicologic pathology* **23**, 349-355 (1995).
22. Ökva, K., *et al.* Refinements for intragastric gavage in rats. *Scandinavian Journal of Laboratory Animal Sciences* **33**, 243-252 (2006).
23. Eichenbaum, G., *et al.* Impact of gavage dosing procedure and gastric content on adverse respiratory effects and mortality in rat toxicity studies. *Journal of Applied Toxicology* **31**, 342-354 (2011).
24. Damsch, S., *et al.* Gavage-related reflux in rats: identification, pathogenesis, and toxicological implications. *Toxicologic pathology* **39**, 348-360 (2011).
25. Vandenberg, L.N., Welshons, W.V., Vom Saal, F.S., Toutain, P.L. & Myers, J.P. Should oral gavage be abandoned in toxicity testing of endocrine disruptors? *Environ Health* **13**, 46 (2014).
26. Berg, D.J., *et al.* Rapid development of colitis in NSAID-treated IL-10-deficient mice. *Gastroenterology* **123**, 1527-1542 (2002).
27. Kiesler, P., Fuss, I.J. & Strober, W. Experimental Models of Inflammatory Bowel Diseases. *Cell Mol Gastroenterol Hepatol* **1**, 154-170 (2015).
28. Low, D., Nguyen, D.D. & Mizoguchi, E. Animal models of ulcerative colitis and their application in drug research. *Drug Des Devel Ther* **7**, 1341-1357 (2013).
29. McGuckin, M.A., Eri, R., Simms, L.A., Florin, T.H. & Radford-Smith, G. Intestinal barrier dysfunction in inflammatory bowel diseases. *Inflamm Bowel Dis* **15**, 100-113 (2009).
30. Madsen, K.L., *et al.* Interleukin-10 gene-deficient mice develop a primary intestinal permeability defect in response to enteric microflora. *Inflamm Bowel Dis* **5**, 262-270 (1999).
31. Sanidad, K.Z., Xiao, H. & Zhang, G. Triclosan, a common antimicrobial ingredient, on gut microbiota and gut health. *Gut microbes* **10**, 434-437 (2019).
32. Allmyr, M., Panagiotidis, G., Sparve, E., Diczfalusy, U. & Sandborgh-Englund, G. Human exposure to triclosan via toothpaste does not change CYP3A4 activity or plasma concentrations of thyroid hormones. *Basic Clin Pharmacol Toxicol* **105**, 339-344 (2009).
33. Sandborgh-Englund, G., Adolfsson-Erici, M., Odham, G. & Ekstrand, J. Pharmacokinetics of triclosan following oral ingestion in humans. *Journal of toxicology and environmental health. Part A* **69**, 1861-1873 (2006).
34. Rodricks, J.V., Swenberg, J.A., Borzelleca, J.F., Maronpot, R.R. & Shipp, A.M. Triclosan: a critical review of the experimental data and development of margins of safety for consumer products. *Crit Rev Toxicol* **40**, 422-484 (2010).
35. Reagan-Shaw, S., Nihal, M. & Ahmad, N. Dose translation from animal to human studies revisited. *FASEB J* **22**, 659-661 (2008).
36. Verslycke, T., Mayfield, D.B., Tabony, J.A., Capdevielle, M. & Slezak, B. Human health risk assessment of triclosan in land-applied biosolids. *Environmental Toxicology and Chemistry* **35**, 2358-2367 (2016).
37. Lu, F.C. Acceptable daily intake: inception, evolution, and application. *Regul Toxicol Pharmacol* **8**, 45-60 (1988).
38. Yueh, M.F. & Tukey, R.H. Triclosan: A Widespread Environmental Toxicant with Many Biological Effects. *Annu Rev Pharmacol Toxicol* **56**, 251-272 (2016).
39. Weatherly, L.M. & Gosse, J.A. Triclosan exposure, transformation, and human health effects. *J Toxicol Environ Health B Crit Rev* **20**, 447-469 (2017).
40. Wu, J.L., Liu, J. & Cai, Z. Determination of triclosan metabolites by using in-source fragmentation from high-performance liquid chromatography/negative atmospheric pressure chemical ionization ion trap mass spectrometry. *Rapid Commun Mass Spectrom* **24**, 1828-1834 (2010).
41. Poole, A.C., *et al.* Crossover Control Study of the Effect of Personal Care Products Containing Triclosan on the Microbiome. *mSphere* **1**(2016).

42. Pellock, S.J. & Redinbo, M.R. Glucuronides in the gut: Sugar-driven symbioses between microbe and host. *J Biol Chem* **292**, 8569-8576 (2017).
43. Pellock, S.J., *et al.* Gut microbial β -glucuronidase inhibition via catalytic cycle interception. *ACS central science* **4**, 868-879 (2018).
44. Scheinin, T., Butler, D.M., Salway, F., Scallon, B. & Feldmann, M. Validation of the interleukin-10 knockout mouse model of colitis: antitumour necrosis factor-antibodies suppress the progression of colitis. *Clin Exp Immunol* **133**, 38-43 (2003).
45. Berg, D.J., *et al.* Enterocolitis and colon cancer in interleukin-10-deficient mice are associated with aberrant cytokine production and CD4(+) TH1-like responses. *The Journal of clinical investigation* **98**, 1010-1020 (1996).
46. Xie, M., *et al.* Triclocarban exposure exaggerates spontaneous colonic inflammation in Il-10^{-/-} mice. *Toxicol Sci* (2019).
47. Yang, H., *et al.* Triclocarban exposure exaggerates colitis and colon tumorigenesis: roles of gut microbiota involved. *Gut microbes* **12**, 1690364 (2020).
48. Zhang, L. Method for voluntary oral administration of drugs in mice. *STAR Protoc* **2**, 100330 (2021).
49. Cani, P.D., *et al.* Changes in gut microbiota control metabolic endotoxemia-induced inflammation in high-fat diet-induced obesity and diabetes in mice. *Diabetes* **57**, 1470-1481 (2008).
50. Vijay-Kumar, M., *et al.* Metabolic syndrome and altered gut microbiota in mice lacking Toll-like receptor 5. *Science (New York, N.Y.)* **328**, 228-231 (2010).
51. Wang, Y., *et al.* Soluble epoxide hydrolase is an endogenous regulator of obesity-induced intestinal barrier dysfunction and bacterial translocation. *Proc Natl Acad Sci U S A*, 201916189 (2020).
52. Marathe, P.H., *et al.* Effect of altered gastric emptying and gastrointestinal motility on metformin absorption. *Br J Clin Pharmacol* **50**, 325-332 (2000).
53. Singh, B.N. Effects of food on clinical pharmacokinetics. *Clin Pharmacokinet* **37**, 213-255 (1999).
54. Fitzgerald, C.B., *et al.* Comparative analysis of *Faecalibacterium prausnitzii* genomes shows a high level of genome plasticity and warrants separation into new species-level taxa. *BMC Genomics* **19**, 931-931 (2018).
55. Kushinsky, S. & Chen, V.L. The inhibition of β -glucuronidase from bovine liver by 1,4-saccharolactone. *Comparative Biochemistry and Physiology* **20**, 535-542 (1967).
56. Pellock, S.J., *et al.* Three structurally and functionally distinct β -glucuronidases from the human gut microbe *Bacteroides uniformis*. *J Biol Chem* **293**, 18559-18573 (2018).
57. Rasmussen, T.S., *et al.* Synthesis of uronic-Noeurostegine – a potent bacterial β -glucuronidase inhibitor. *Organic & Biomolecular Chemistry* **9**, 7807-7813 (2011).
58. Bhatt, A.P., *et al.* Targeted inhibition of gut bacterial β -glucuronidase activity enhances anticancer drug efficacy. *Proceedings of the National Academy of Sciences* **117**, 7374 (2020).
59. Morgan, M.K. & Clifton, M.S. Dietary Exposures and Intake Doses to Bisphenol A and Triclosan in 188 Duplicate-Single Solid Food Items Consumed by US Adults. *Int J Environ Res Public Health* **18**(2021).

REVIEWERS' COMMENTS

Reviewer #2 (Remarks to the Author):

The authors have addressed and answered all of my concerns and comments.

Reviewer #3 (Remarks to the Author):

The authors addressed very thoughtfully the concerns I raised. I have no other comments.

Response to reviewers

Reviewer #2:

The authors have addressed and answered all of my concerns and comments.

We would like to thank the reviewer for the positive comment.

Reviewer #3:

The authors addressed very thoughtfully the concerns I raised. I have no other comments.

We would like to thank the reviewer for the positive comment.

Other reviewer comment:

I understand the authors mean that the presence of free TCS in the colon of GF mice may be the result of host digestive enzyme metabolism of TCS-G to TCS. However, the use of “likely derived from the small intestine digesta”, which seems a bit unclear and obscure. Do they mean that TCS-G metabolism is likely a process of both host and bacterial enzymes? Or that free TCS is ingested as such from the food? Or both?

We think that the free TCS in the colon of germ-free mice is ingested as such from the food. After mice were exposed to 80 ppm TCS in diet, part of the ingested TCS remains unchanged in small intestine (as supported by the presence of free TCS in the digesta of small intestine in Fig. 1a), then these free TCS could enter the colon.

Based on reviewer's comment, we revised the manuscript: “We observed the presence of free TCS in the colon of germ-free mice (Fig. 2h), and this could be from ingested TCS from the food: we showed that after mice were exposed to 80 ppm TCS in diet, part of the ingested TCS remained unchanged in the small intestine as free TCS was detected in the digesta of small intestine (Fig. 1a). This could also happen in the germ-free mice and the free TCS in the small intestine could then enter the colon with the flow of digesta”.